# INTERPOLATING COMPRESSED PARAMETER SUBSPACES

## ABSTRACT

Though distribution shifts have caused growing concern for machine learning scalability, solutions tend to specialize towards a specific type of distribution shift. Methods for label shift may not succeed against domain or task shift, and vice versa. We learn that constructing a *Compressed Parameter Subspaces (CPS)*, a geometric structure representing distance-regularized parameters mapped to a set of train-time distributions, can maximize average accuracy over a broad range of distribution shifts concurrently. We show sampling parameters within a CPS can mitigate backdoor, adversarial, permutation, stylization and rotation perturbations. We also show training a hypernetwork representing a CPS can adapt to seen tasks as well as unseen interpolated tasks.

## 1 INTRODUCTION

Suppose your model is expecting distribution shift at test-time. Is it task shift? Domain shift? Rotations? What if it's an adversarial attack, or even a backdoor attack? Each strand of distribution shift research has provided robust and adaptive techniques against their respective types of shift, and achieved state-of-the-art results. In this work, we take a step towards a method that can adapt to multiple types of distribution shift concurrently.

Recent work on the geometry of the loss landscape, such as neural subspaces (Wortsman et al., 2021) and mode connectivity (Fort & Jastrzebski, 2019; Draxler et al., 2019; Garipov et al., 2018) discovered properties of robustness between multiple parameters. Departing from constructing subspaces w.r.t. a single/unperturbed input distribution, we investigate the construction of subspces w.r.t. multiple perturbed distributions, and find improved mappability between shifted distributions and low-loss parameters contained in these subspaces.

**Contributions.** We share a method to construct a compressed parameter subspace such that the likelihood of a parameter sampled from this subspace can be mapped to a shifted input distribution is higher. We demonstrate a high average accuracy across distribution shifts in single and multiple test-time settings (Figure 1). We show improved robustness across perturbation types, reduced catastrophic forgetting on Split-CIFAR10/100, and strong capacity for multi-task solutions and unseen/distant tasks.

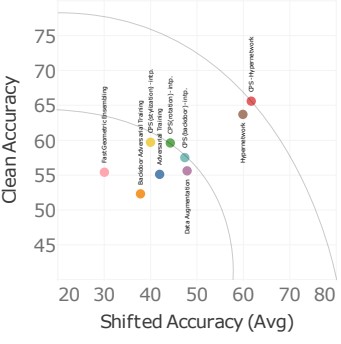

Figure 1: *Shift-optimality:* We plot the trade-off frontier between methods performing across a range of distribution shift types. For single test-time distributions (Table 1), Clean Accuracy is the accuracy w.r.t. Clean Test Set; Shifted Accuracy is the average accuracy across different shifts (Backdoor/Adversarial Attack, Random Permutations, Stylization, Rotation). For multiple test-time distributions (Table 2), Clean Accuracy is the average accuracy of the hypernetwork evaluated after each task; Shifted Accuracy is the average accuracy of the hypernetwork evaluated after the last task.

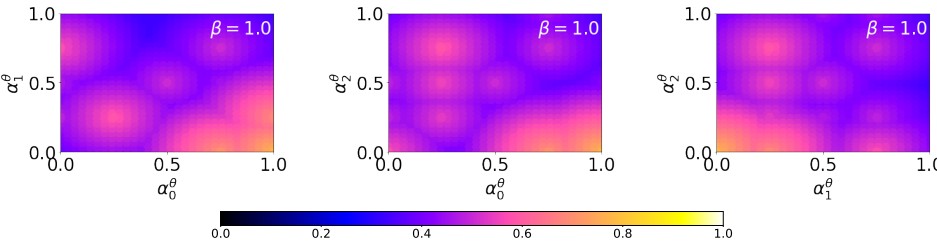

Figure 2: *Change in parameter subspace dynamics:* Landscape of unique lowest-loss parameters mapped to interpolated inputs. Refer to Appendix A.3.3 for supplementary visualization.

## 2 PROBLEM: ADAPTATION TO DISTRIBUTION SHIFT

We first motivate the problem of adapting to distribution shifts from literature. Then we formulate the different types of distribution shift into a generalized notation and problem setup.

### 2.1 RELATED WORK

Different aspects of distribution shift have been studied individually and methods of robustness/adaptation have been proposed. More recently, there is work studying what happens when deep learning architectures encounter test-time inputs manifesting multiple types of distribution shifts. Qi et al. (2021) and Datta (2022) combine adversarial attacks with stylization and domain shift Liu et al. (2021) combined adversarial perturbations with backdoor trigger perturbations to craft stealthy triggers to perform backdoor attacks. However, methods of robustness/adaptation for a single test-time distribution (e.g. adversarial/backdoor attacks) do not strongly overlap with that for multiple test-time distributions (e.g. domain adaptation, meta/continual learning). Hence, these methods generally require the distribution shift to be specified at train-time. For example, Weng et al. (2020) found a trade-off between adversarial defenses optimized towards adversarial perturbations against backdoor defenses optimized towards backdoor perturbations, where the adoption of one defense worsened robustness to the other's attack. Though data augmentation (Yun et al., 2019b; Borgnia et al., 2020) and adversarial training (Goodfellow et al., 2015; Geiping et al., 2021) as a technique can mitigate adversarial attacks and backdoor attacks, they need to be trained on one of the perturbation types to manifest robustness. To retain adversarial robustness in few-shot learning, Goldblum et al. (2020) and Wang et al. (2021) investigated methods of integrating adversarial training into MAML. As such, we are motivated to contribute a method of adaptation/robustness that can multiple types of distribution shifts concurrently.

### 2.2 PRELIMINARIES

Let $\mathcal{X}, \mathcal{Y}, \mathcal{K}$ be denoted as the input space, coarse label space, and fine label space respectively. Coarse labels are the higher-order labels of fine labels. A base learner function $f$ with parameters $\theta$ accepts inputs $\mathbf{x}$ to return predicted labels $\bar{\mathbf{y}} = f(\theta; \mathbf{x})$. $\theta$ is computed such that it minimizes the loss between the ground-truth and predicted labels: $\mathcal{L}(\theta; \mathbf{x}, \mathbf{y}) = \frac{1}{|\mathbf{x}|} \sum_i^{|\mathbf{x}|} (f(\theta; \mathbf{x}) - \mathbf{y})^2$.

**Definition 1.** *A **distribution shift** is the divergence between a train-time distribution $\mathbf{x_0}, \mathbf{y_0}$ and a test-time distribution $\hat{\mathbf{x}}, \hat{\mathbf{y}}$, where $\hat{\mathbf{x}}, \hat{\mathbf{y}}$ is an interpolated distribution between $\mathbf{x_0}, \mathbf{y_0}$ and a target distribution $\mathbf{x_i^\Delta}, \mathbf{y_i^\Delta}$ such that $\hat{\mathbf{x}} = \sum_i^N \alpha_i \mathbf{x_i^\Delta}$ and $\hat{\mathbf{y}} = \mathbf{y_i}$ where $i = \arg\max_i \alpha_i$.*

> **Definition 1.1** A **disjoint distribution shift** is a distribution shift of one target distribution $|\{\alpha_i\}| = 2$ such that $\hat{\mathbf{x}} = \alpha \mathbf{x^\Delta} + (1 - \alpha)\mathbf{x_0}$.
>
> **Definition 1.2** A **joint distribution shift** is a distribution shift of multiple target distributions $|\{\alpha_i\}| > 2$ such that $\hat{\mathbf{x}} = \sum_i^N \alpha_i \mathbf{x_i^\Delta}$.

For $N$ distributions $\{\mathbf{x_0} \mapsto \mathbf{y_0}, \mathbf{x_1^\Delta} \mapsto \mathbf{y_1^\Delta}, ..., \mathbf{x_N^\Delta} \mapsto \mathbf{y_N^\Delta}\}$ containing $(N-1)$ target distributions, we sample interpolation coefficients $\alpha_i \sim [0,1]$ s.t. $\alpha_i \mapsto \mathbf{x_i}$ and $\sum_i^N \alpha_i \leq N$. In CPS training, $N$ is the number of train-time distributions used in training the subspace; in task interpolation, $N$

is the number of train/test-time distributions used in interpolating between task sets. Given a set of valid interpolated input distributions (manifesting disjoint/joint shifts) and their corresponding interpolation coefficients $\{\{\alpha_i\} \mapsto \hat{\mathbf{x}}\}$, the primary objective of this work is to compute the set of parameters $\{\theta\}$ that can be mapped to shifted distributions $\hat{\mathbf{x}} \mapsto \theta$ such that we minimize the cumulative loss $\sum_{\theta,\{\alpha_i\}}^{\{\theta\},\{\{\alpha_i\}\}} \mathcal{L}(\theta; \sum_i^{|\{\alpha_i\}|} \alpha_i \mathbf{x_i}, \hat{\mathbf{y}})$.

# 3 COMPRESSED PARAMETER SUBSPACES (CPS)

## 3.1 SUBSPACE CONSTRUCTION

The CPS is constructed by training multiple parameters in parallel, where each parameter is trained in a different, shifted distribution. Parameter point-estimates can be interpolated between the trained parameters (which set the bounds of the subspace). In this section, we precisely define the method, subsequently explain why we expect low-loss parameters to reside in this subspace that can be mapped to shifted distributions in the input space, then finally share the inference techniques used.

| Algorithm 1: CPS: Train | Algorithm 2: CPS: Evaluate |
|---|---|
| **trainSpace** $(f, \{\theta_i\}^N, \{D_i\}^N, \beta, E)$ | **evalSpace** $x, \{\theta_i\}^N, \{\alpha_i\}^N$ |
|   **Input** : Model $f$, Parameters $\{\theta_i\}^N$, Train set $\{D_i\}^N$, Distance coefficient $\beta$, Epochs $E$ |   **Input** : Input $x$, Trained parameters $\{\theta_i\}^N$, Interpolation coefficients $\alpha_i \sim [0,1]$ s.t. $\sum_i^N \alpha_i \leq N$ |
|   **Output** : Trained parameters $\{\theta_i\}^N$ |   **Output** : Predicted output $\hat{y}$ |
|   Constant initialization $\{\theta_{\text{init}} \leftarrow \theta_i\}^N$ |   $\theta \leftarrow \sum_i^N \alpha_i \theta_i$ |
|   Train each parameter against their indexed train set in parallel per epoch |    $\hat{y} \leftarrow f(x, \theta)$ |
|   **foreach** epoch $e \in E$ **do** |    **return** $\hat{y}$ |
|    **foreach** $\theta_i, D_i \in \{\theta_i\}^N, \{D_i\}^N$ **do** | |
|     **foreach** batch $x, y \in D_i$ **do** | |
|      $\hat{y} \leftarrow f(x, \theta_i)$ | |
|      $\mathcal{L} \leftarrow \ell(\hat{y}, y) + \beta\,\mathsf{cos}(\{\theta_i\}^N)$ | |
|      Backprop $\theta_i$ w.r.t. $\mathcal{L}$ | |
|   **return** $\{\theta_i\}^N$ | |

**Definition 2.** *The **parameter space** $\Theta \in \mathbb{R}^M$ is an $M$-dimensional space where parameter point-estimates $\theta \sim \Theta$ are sampled and loaded into a base learner function $f(\theta; \cdot)$. A **parameter subspace** $\vartheta$ is a bounded subspace contained in $\Theta$ such that:*

$$\vartheta = \Big\{ \sum_i^N \alpha_i \theta_i \in \Theta \,\Big|\, 0 \leq \alpha_i \leq 1, 0 \leq \sum_i^N \alpha_i \leq N, \{\min \mathcal{L}(\theta_i; X_i, Y_i)\}^{i \in N} \Big\}$$

**Definition 2.1** *A **compressed parameter subspace** is a parameter subspace with the constraint that the distance between the set of end-point parameters should be minimized (Algorithm 1) such that:*

$$\vartheta = \Big\{ \sum_i^N \alpha_i \theta_i \in \Theta \,\Big|\, 0 \leq \alpha_i \leq 1, 0 \leq \sum_i^N \alpha_i \leq N, \{\min \mathcal{L}(\theta_i; X_i, Y_i)\}^{i \in N}, \boxed{\min \mathsf{dist}(\{\theta_i\}^N)} \Big\}$$

From Theorem 2, we show that the end-point parameters converge towards a local region in the parameter space; as the average distance between any end-point and the centre of these points is lower upon loss convergence, this space is referred to as being *compressed*. From Theorem 1, we show that the end-point parameters (and subsequently interpolated parameters within the subspace) contain real-valued artifacts of opposing end-point parameters. Further, from Theorem 3, we show that an interpolated input set $\hat{\mathbf{x}}$ has a lower expected distance towards any interpolated (or end-point) parameter point-estimate in a compressed space than an uncompressed space. From these properties, we empirically validate that interpolated input sets in single and multiple test-time distributions can be mapped back to interpolated parameters in the CPS.

**Theorem 1.** *For $N + 1$ parameters trained in parallel, any individual parameter $\theta_i$ is a function of its mapped data $X_i \mapsto Y_i$ as well as the remaining parameters $\{\theta_n\}^{n \in N}$ (and by extension, $X_n \mapsto Y_n$). Hence, a linearly-interpolated parameter between $\theta_i$ and $\{\theta_n\}^{n \in N}$ is a function of $\{\theta_n\}^{n \in N}$ and $\theta_i | \{\theta_n\}^{n \in N}$.*

*Proof sketch.* Denoting $\mathcal{C} = \sum_n^N \theta_n$, computing $\mathcal{L}(\theta_i) = \mathcal{L}(\theta_i, X_i, Y_i) + \texttt{dist}(\theta_i, \{\theta_n\}^{n \in N})$ such that $\mathcal{L}(\theta_i) = 0$ leads to the property that:

$$\theta_i := \frac{2\mathcal{C} - X_i \pm \sqrt{-4X_i\mathcal{C} + 4NY_i + X_i^2}}{2N}$$

This and subsequent results find that any end-point and interpolated parameter $\theta_i$ will be a function of a nested $\{\theta_n\}^{n \in N}$. Refer to proof in Appendix A.1.

**Theorem 2.** *For $N + 1$ parameters trained in parallel, for an individual parameter $\theta_i$ and another parameter $\theta_n$ where $n \in N$, the change in loss per iteration of SGD with respect to each parameter $\frac{\partial \mathcal{L}(\theta_i; X_i)}{\partial \theta_i}, \frac{\partial \mathcal{L}(\theta_n; X_n)}{\partial \theta_n}$ will both be in the same direction (i.e. identical signs), and will tend towards a local region of the parameter space upon convergence of loss.*

*Proof sketch.* By computing the incremental loss attributed to the distance regularization term upon convergence $\Delta \mathcal{L}(\theta_i)_e = 0$, we find that:

$$\frac{\partial \mathcal{L}(\theta_i; X_i)}{\partial \theta_i} \cdot \frac{\partial \mathcal{L}(\theta_n; X_n)}{\partial \theta_n} \equiv \frac{1}{2}\left[\left(\frac{\partial \mathcal{L}(\theta_n; X_n)}{\partial \theta_n}\right)^2 + \left(\frac{\partial \mathcal{L}(\theta_i; X_i)}{\partial \theta_i}\right)^2\right]$$

Subsequently, we find the respective cumulative loss of (and consequently distance between) $\theta_i$ and $\theta_n$ from iterations $e = 0$ to $e = E - 1$:

$$\left|\sum_e^{E-1} \frac{\partial \mathcal{L}(\theta_{i,t=e}; X_i)}{\partial \theta_{i,t=e}}\right| - \left|\sum_e^{E-1} \frac{\partial \mathcal{L}(\theta_{n,t=e}; X_n)}{\partial \theta_{n,t=e}}\right|$$

to be lower when $\frac{\partial \mathcal{L}(\theta_i; X_i)}{\partial \theta_i} \cdot \frac{\partial \mathcal{L}(\theta_n; X_n)}{\partial \theta_n} > 0$ (i.e. active distance regularization) than when $\frac{\partial \mathcal{L}(\theta_i; X_i)}{\partial \theta_i} \cdot \frac{\partial \mathcal{L}(\theta_n; X_n)}{\partial \theta_n} \leq 0$. Refer to proof in Appendix A.1.

**Theorem 3.** *For $N + 1$ parameters trained in parallel, the expected distance between the optimal parameter of an interpolated input $\hat{\mathbf{x}} \mapsto \hat{\theta}$ would be smaller with respect to sampled points in a distance-regularized subspace than non-distance-regularized subspace.*

*Proof sketch.* For the the ground-truth parameter of a sampled interpolated input distribution, in order for the loss to converge $\mathcal{L}(\hat{\theta}) \to 0$, the product of the loss terms converges in the same direction $\frac{\partial \mathcal{L}(\hat{\theta}; X_i)}{\partial \hat{\theta}} \cdot \frac{\partial \mathcal{L}(\hat{\theta}; X_n)}{\partial \hat{\theta}} > 0$. For iterations $e = 0$ to $e = E - 1$, the cumulative loss (and consequently distance) between the ground-truth parameter and parameter end-points

$$\mathbb{E}\left[\left|2\left[\alpha_i \frac{\partial \mathcal{L}(\hat{\theta}_{t=e}; \mathbf{x_i})}{\partial \hat{\theta}_{t=e}} + \alpha_n \frac{\partial \mathcal{L}(\hat{\theta}_{t=e}; \mathbf{x_n})}{\partial \hat{\theta}_{t=e}}\right] + \left[\gamma_n \frac{\partial \mathcal{L}(\theta_{n,t=e}; \mathbf{x_n})}{\partial \theta_{n,t=e}} - \gamma_i \frac{\partial \mathcal{L}(\theta_{i,t=e}; \mathbf{x_i})}{\partial \theta_{i,t=e}}\right]\right|\right]$$

is closer in the distance-regularized space than non-distance-regularized space. Refer to proof in Appendix A.1.

Instead of a single distribution with multiple random initializations, we begin with a set of shifted distributions $\{\mathbf{x_i}\}^N$ and a single constant (randomized) initialization. For a single test-time distribution (CIFAR10), we slice subsets of the train set and insert unique perturbations of a given type (backdoor / stylization / rotation) into each subset. For multiple test-time distributions (CIFAR100), we use different task sets as each distribution. Each model is a $\ell$-layer CNN trained with early-stopping at loss 1.0 (to accommodate computational load for training, hence limiting accuracy). We load the $N$ sets in parallel. At the end of each epoch, we compute the loss of each model's parameters with respect to their train-time sets. We then compute the average cosine distance between each model's weight against all the other model's weights. We add this distance multiplied by a distance coefficient $\beta = 1.0$ to the total loss, and update each model's weights with respect to this total loss.

## 3.2 SUBSPACE INFERENCE

There are capacity-efficient methods to query parameters from the CPS without explicitly/separately storing large subsets of the CPS. We list the following three classes of test-time inference with a subspace.

### 3.2.1 UNBIASED SAMPLING

In this sampling scheme, we do not use any heuristics in selecting parameters.

- We can compute predictions $\bar{y} = \ell(\theta^*; \hat{\mathbf{x}})$ given an input and the **centre** parameter point-estimate $\theta^* = \sum_i^N \frac{1}{N} \theta_i$ (the average of the end-point parameters).

- We can compute a mean prediction $\bar{y} = \frac{1}{M} \sum_j^M \ell(\theta_j^*; \hat{\mathbf{x}})$ given an input and an **ensemble** of $M$ randomly-sampled parameter point-estimates in the subspace; we randomly sample $M = 1000$ interpolation coefficients $\{\alpha_{j,i}\}^{M \times N}$ to return an ensemble set of parameter point estimates $\{\theta_j^*\}^M = \{\alpha_{j,i}\}^{M \times N} \cdot \{\theta_i\}^N$.

### 3.2.2 GUIDED SAMPLING

Guided sampling makes use of a heuristic in the selection of parameters to be used for inference.

- We can compute predictions $\bar{y} = \ell(\theta_{j^*}; \hat{\mathbf{x}})$ given an input and a lowest-loss **interpolated** parameter point-estimate, where $j^* := \arg\min_{j \sim M} \mathcal{L}(\theta_j; \mathbf{x}, \mathbf{y})$. This is specifically a unique-task solution, where the interpolated parameter maps back to one task/distribution $\hat{\mathbf{x}} \mapsto \theta_{j^*}$. For a multi-task solution, an interpolated parameter is mapped back to a set of $T$ task/distributions $\{\hat{\mathbf{x}}_t\}^T \mapsto \theta_{j^*}$, where $j^* := \arg\min_{j \sim M} \sum_t^T \mathcal{L}(\theta_j; \mathbf{x_t}, \mathbf{y_t})$. To identify this interpolated parameter that returns the lowest-loss (maximum-accuracy) for a given test-time distribution, we can actively and iteratively sample parameters with respect to the centre of the subspace, or linearly enumerate through interpolated parameters between the known boundary parameters. An example of a scheme would be, given K-shots, we can sample parameters and use the parameter of lowest-loss for the task. This could also be extended to construct an ensemble. In this work, we search for the lowest-loss parameter to show that a compressed parameter subspace has a higher likelihood of containing a shift-optimal parameters.

- We can compute predictions $\bar{y} = \ell(\theta_{i^*}; \hat{\mathbf{x}})$ given an input and lowest-loss **boundary** parameter, where the latter is the parameter from a set of $N$ boundary or end-point parameters that returns the lowest loss $i^* := \arg\min_{i \sim N} \mathcal{L}(\theta_i; \mathbf{x}, \mathbf{y})$. The boundary parameter may be used if the task index is known. In this work, we use the boundary parameter as an ablation, and to check if the lowest-loss interpolated parameter is simply returning a boudary parameter or not.

### 3.2.3 GENERALIZED FUNCTION

We can generalize or represent the subspace implicitly with a learning function. We evaluate CPS-regularization in hypernetworks, an existing continual learning strategy. In-line with Zenke et al. (2017) and von Oswald et al. (2020), we adopt the Split CIFAR10/100 benchmark, where we train on CIFAR10 first, then train sequentially on the next 5 sets of 10-label CIFAR100. All tasks share the same 10 coarse labels.

A **hypernetwork** $\hbar(\mathbf{x}, I) = \ell(\mathbf{x}; m\ell(\theta_{m\ell}; I))$ is a pair of learners, the base learner $\ell : \mathcal{X} \mapsto \mathcal{Y}$ and meta learner $m\ell : \mathcal{I} \mapsto \Theta_\ell$, such that for the conditioning input $I$ of input $\mathbf{x}$ (where $\mathcal{X} \mapsto \mathcal{I}$), $m\ell$ produces the base learner parameters $\theta_I = m\ell(\theta_{m\ell}; I)$. The function $m\ell(\theta_{m\ell}; I)$ takes a conditioning input $I$ to returns parameters $\theta_I \in \Theta_\ell$ for $\ell$. The meta learner parameters and each base learner parameters reside in their distinct parameter spaces $\theta_{m\ell} \in \Theta_{m\ell}$ and $\theta_\ell \in \Theta_\ell$. The learner $\ell$ takes an input $\mathbf{x}$ and returns an output $\bar{y} = \ell(\theta_I; \mathbf{x})$ that depends on both $\mathbf{x}$ and the task-specific input $I$. $T$ is number of tasks, $t$ is index of specific task being evaluated ($t^*$ being current task), $\omega$ is task regularizer. $\theta_{m\ell}$ is the hypernetwork parameters at the current task's training timestep, $\theta_{m\ell}^*$ is the hypernetwork parameters before attempting to learn task $t^*$, and $\Delta\theta_{m\ell}$ is the candidate parameter change computed by the optimizer (Adam). We do not use any separate conditioning input (i.e. task embeddings), and use the test-time input as the primary argument, i.e. $\mathcal{I} \equiv \mathcal{X}$.

$$
\begin{aligned}
\mathcal{L}_{m\ell} = {} & \mathcal{L}_{t^*}(m\ell(\theta_{m\ell}; \mathbf{x_{t^*}}); \mathbf{x_{t^*}}, \mathbf{y_{t^*}}) \\
& + \frac{\omega}{t^* - 1} \sum_t^{t^*-1} \left\| \ell(m\ell(\theta_{m\ell}^*; \mathbf{x_t}); \mathbf{x_t}, \mathbf{y_t}) - \ell(m\ell(\theta_{m\ell} + \Delta\theta_{m\ell}; \mathbf{x_t}); \mathbf{x_t}, \mathbf{y_t}) \right\|^2 \\
& + \boxed{\frac{\beta}{t^* - 1} \sum_t^{t^*-1} \texttt{dist}(\theta_{m\ell}, \theta_{m\ell,t})}
\end{aligned}
\tag{1}
$$

Starting from a random initialization, we train the hypernetwork first on CIFAR10 (interpretable as re-initializing the network with pre-trained weights, which findings in Neyshabur et al. (2020) may indicate that subsequent task parameters may reside in a shared low-loss basin), and store these parameters $\theta_{m\ell,t}$ in a parameter set $\{\theta_{m\ell,t}\}^T$. Then we train on a subsequent CIFAR100 task with the Eqt. 1 loss function, where we compute loss w.r.t. the inputs and current timestep's parameters, change in loss w.r.t. a previous task between using the proposed parameters and the parameters last updated at that task (stored in the parameter set; this is enumerated for all past tasks in sequence), and the distance between the current parameters and all prior parameters. Unlike our prior (multi-task) implementation, (i) we minimize the cosine distance between a current task's parameters against prior parameters sequentially, not in parallel (i.e. the subspace end-points are sequentially fixed in the parameter space, we cannot dynamically move the subspace towards a different region), (ii) we do not gain visibility to all task parameters at once, and (iii) we are computing distance w.r.t. multi-task parameters (i.e. each task's parameters is applicable to its own and prior tasks).

# 4 METHODOLOGY

We summarize our data processing and baseline methods, with additional details in Appendix A.2.

**Test-time distributions.** We evaluate on CIFAR10 and CIFAR100 (Krizhevsky, 2009) for single (non-task-shift) and multiple test-time distributions (task-shift) respectively. For label shift w.r.t. train-time perturbations, we implement backdoor attacks with Random-BadNet (Datta & Shadbolt, 2022), a variation of BadNet (Gu et al., 2019) to allow for multiple trigger patterns. For label shift w.r.t. test-time perturbations, we implement adversarial attacks for targeted label shift with Projected Gradient Descent (Madry et al., 2018), and implement random permutations for untargeted label shift with Random-BadNet (in-line with PermuteCIFAR10). We implement stylization with Adaptive Instance Normalization (Huang & Belongie, 2017). We rotate images in-line with RotateCIFAR10. For task shift, where the tasks are pre-defined in the dataset with shared or non-shared label sets, we evaluate on both the end-point tasks as well as interpolated tasks.

**Task interpolation.** Interpolation is an existing data/task augmentation technique. MLTI (Yao et al., 2021) linearly interpolates the hidden state representations at the $\ell$th layer of a model when a set of tasks are passed through it. Mixup (Zhang et al., 2017b) linearly interpolates raw inputs and labels for data augmentation. We evaluate interpolated tasks by linearly interpolating a set of tasks: for a set of tasks $\{X\}^N$ where each task $X$ is a set of inputs $x$ (and the number of inputs per task is identical, i.e. $\{\{x\}^M\}^N$), we linearly-interpolate between indexed inputs across a task set to return an interpolated task: $\hat{\mathbf{x}} = \sum_i^N \sum_j^M \alpha_i x_{i,j}$. In this work, we only interpolate between task sets with identical labels (label-shared).

**Baselines.** We baseline with CutMix data augmentation (Yun et al., 2019a), adversarial training with PGD (Goodfellow et al., 2015), and backdoor adversarial training (Geiping et al., 2021). We additionally evaluate a subspace and ensembling method, Fast Geometric Ensembles (Garipov et al., 2018). In a continual learning setting, we evaluate with hypernetworks (von Oswald et al., 2020).

# 5 EVALUATION

We summarize below the experiments evaluated (detailed configurations in Appendix A.2).

- **Change in parameter subspace dynamics:** We plot the changes in loss and cosine distance for training CPS with 3 train-time distributions *(Figure 2, Appendix A.3.1 )*. We linearly-interpolate between tasks and end-point parameters *(Figure 2)*. We plot the individual tasks along interpolation coefficients $\{\alpha_i^X\}^N$, and unique lowest-loss parameters that can correspond to a task along interpolation coefficients $\{\alpha_i^\theta\}^N$. We also render an alternative continuous cross-sectional plot *(Appendix A.3.3)*. The task end-points in these plots are: 3 tasks, 3 × same coarse label, train-time task set.

- **CPS vs single test-time distributions:** We evaluate baselines and CPS trained on 3 subsets of perturbed CIFAR10 against a range of test-time perturbations *(Table 1)*. We evaluate the relationship between CPS, number of perturbed sets trained on, and model capacity *(Appendix A.3.4)*.

- **CPS vs multiple test-time distributions:** We evaluate the insertion of a CPS-regularization term in training hypernetworks for continual learning *(Table 3)*. We evaluate the improvement in mapping

low-loss parameters in CPS *(Table 2)*. We evaluate CPS (varying for over-parameterization w.r.t. number of tasks and model capacity) against seen/unseen tasks (varying for additional perturbation types) *(Appendix A.3.8)*. We also evaluate varying model width against train-time distribution label set diversity *(Appendix A.3.6)*. We also evaluate varying model depth against varying distinct coarse label sets in train-time distributions *(Appendix A.3.5)*. We also evaluate varying model depth against varying task label set diversity (different fine and coarse labels) *(Appendix A.3.7)*.

## 5.1 SINGLE TEST-TIME DISTRIBUTION

**(Observation 1)** *A parameter subspace can be found where the distance between end-point parameters to the centre is reduced.* The intended result from cosine distance minimization is not to converge cosine distance to 0 (e.g. using a larger capacity model can result in lower cosine distance between parameters); the intended result is to minimize the distance between parameters compared to without regularization, and through this process find an interplotable subspace in the parameter space. Comparing the distance w.r.t. opposing end-point parameters and the centre in Figure 2 and Appendix A.3.1, we find that the distances have comparatively converged when $\beta = 1$ than $\beta = 0$ for single test-time distributions, but the convergence in distance is not as apparent for multiple test-time distributions; the latter may suggest a minimum distance (interference) is required between certain tasks particularly if there is limited transferable features between them.

From Figure 2 and Appendix A.3.1, when training w.r.t. a single test-time distribution, we observe that the cosine distance (w.r.t. subspace centre) first increases substantially before decreasing and continuing to stay at a low level. As opposed to SGD performing point-estimate optimization, this phenomenon can be interpreted as SGD performing subspace optimization, in-line with Theorem 2. Without distance regularization, the trajectory taken by SGD tends to depend more on the initialization point (Fort et al., 2020). In-line with Neyshabur et al. (2020), the parameters at peak cosine distance may be interpreted as a second, albeit unshared, pre-trained weight initialization that propagate parameters towards a same low-loss basin.

**(Observation 2)** *Sampling parameter point-estimates in the CPS can attain a high average robustness accuracy across different perturbations or shifts.* Evaluated against the clean test-time distribution, we find that sampling point-estimates in the CPS can retain its accuracy across varying perturbation types (Table 1, Appendix A.3.8). For a single test-time distribution (CIFAR10) with varying perturbation types, we find that inference with the centre or ensemble can yield a higher average accuracy across perturbed test-time distributions than wide-spectrum and/or niche defenses, particularly CPS trained on backdoored subsets. We find that the increased low-loss parameter mappings to the input space enables robustness for multiple perturbation types, in-line with Theorem 3. Though outperforming on average, CPS underperforms data augmentation with respect to test-time stylization and rotation. Though a lowest-loss point-estimate can be found for most perturbation types, we note that a comparably-accurate one cannot be found for test-time stylization. Though CPS trained on rotated sets can contain lowest-loss point-estimates for rotational perturbations, an averaging strategy (neither averaging the parameters to compute the centre, nor ensembling the predictions of 1000 parameters) does not work as well as other perturbation types. We achieve above-chance (CIFAR10 10%, CIFAR100 20%) performance on stylization and rotation, and show that a lowest-loss parameter can be interpolated within the CPS for these cases. Other than a backdoor attack, training a CPS on any perturbed type of subset can attain similar performance.

## 5.2 MULTIPLE TEST-TIME DISTRIBUTIONS

**(Observation 3)** *Constructing a CPS can yield a subspace containing low-loss parameters for corresponding inputs.* Based on Table 2, compared to a non-CPS, we observe that more points interpolated in the input space can be mapped to low-loss interpolated parameters in the parameter space. We would expect a larger subspace to contain more low-loss parameters, yet counter-intuitively a subspace compressed w.r.t. opposing boundary parameters contains more mappable low-loss parameters (Theorem 3). We also note that the interpolated points perform better than reusing boundary parameters. This result is visualized in Appendix A.3.3 (and Figure 2): more points in the input space can be mapped with higher accuracy, and fewer, sparser yet more-accurate points can be located in the parameter space that can map back to these inputs. In addition to these interpolated parameters that can be mapped back to shifted distributions, the average (ensemble) accuracy of the

| | | Acc w.r.t. seen distributions | | | Acc w.r.t. unseen shift distributions | | |
|---|---|---|---|---|---|---|---|
| | | Clean Test Set | Backdoor Attack | Adversarial Attack | Random Permutations | Stylization | Rotation |
| Data Augmentation | | 55.6 | $49.0 \pm 25.2$ | 40.4 | $49.6 \pm 23.2$ | $49.0 \pm 21.3$ | $51.4 \pm 23.7$ |
| Adversarial Training | | 55.1 | $37.2 \pm 22.4$ | 50.0 | $41.8 \pm 22.4$ | $40.0 \pm 19.9$ | $40.8 \pm 23.3$ |
| Backdoor Adversarial Training | | 52.3 | $45.0 \pm 23.5$ | 31.2 | $36.2 \pm 21.0$ | $37.6 \pm 21.5$ | $39.2 \pm 20.5$ |
| Fast Geometric Ensembling | | 55.4 | $32.0 \pm 11.9$ | 32.5 | $29.2 \pm 18.7$ | $25.2 \pm 18.6$ | $31.4 \pm 18.5$ |
| CPS (backdoor) | centre | $55.3 \pm 15.8$ | $55.3 \pm 15.8$ | $47.7 \pm 2.5$ | $54.7 \pm 21.7$ | $23.9 \pm 4.0$ | $28.2 \pm 19.8$ |
| | ens. (mean) | $55.3 \pm 15.7$ | $55.3 \pm 15.7$ | $47.3 \pm 2.2$ | $54.2 \pm 20.8$ | $23.5 \pm 3.6$ | $28.2 \pm 19.1$ |
| | intp. (max) | $57.5 \pm 15.5$ | $57.5 \pm 15.5$ | $60.5 \pm 1.2$ | $61.7 \pm 19.8$ | $27.0 \pm 3.8$ | $30.2 \pm 19.1$ |
| CPS (stylization) | centre | $57.5 \pm 17.8$ | $21.2 \pm 1.4$ | $50.6 \pm 2.1$ | $50.3 \pm 19.4$ | $24.2 \pm 2.4$ | $26.1 \pm 19.5$ |
| | ens. (mean) | $57.3 \pm 18.3$ | $20.8 \pm 1.4$ | $50.4 \pm 2.3$ | $59.2 \pm 19.3$ | $24.1 \pm 2.8$ | $26.1 \pm 19.5$ |
| | intp. (max) | $59.7 \pm 18.2$ | $22.7 \pm 1.6$ | $60.7 \pm 1.8$ | $60.1 \pm 19.4$ | $27.4 \pm 3.4$ | $29.1 \pm 19.2$ |
| CPS (rotation) | centre | $19.0 \pm 1.8$ | $18.1 \pm 6.1$ | $19.0 \pm 1.4$ | $19.3 \pm 12.7$ | $13.1 \pm 1.1$ | $27.4 \pm 20.3$ |
| | ens. (mean) | $19.2 \pm 1.8$ | $25.3 \pm 9.2$ | $18.1 \pm 1.2$ | $19.2 \pm 11.0$ | $13.1 \pm 1.5$ | $26.9 \pm 19.3$ |
| | intp. (max) | $59.6 \pm 17.2$ | $38.2 \pm 1.2$ | $58.1 \pm 1.6$ | $52.8 \pm 18.9$ | $23.7 \pm 3.1$ | $48.5 \pm 26.6$ |

Table 1: *CPS vs single test-time distributions:* Baselines and CPS trained on 3 subsets of perturbed CIFAR10 (backdoor / stylization / rotation) against varying shifts w.r.t. test-time distributions (mean $\pm$ std w.r.t. number of test-time samples): clean test set (out of 3), backdoor attack (out of 3), adversarial attack (out of 3), random permutations (out of 100), stylization (out of 100), and rotation (out of 100). Base model is 3-layer CNN. The backdoor attack is evaluated w.r.t. clean samples (higher accuracy is higher robustness), not w.r.t. poisoned labels.

| | | Uniform interpolation (of inputs) |
|---|---|---|
| CPS: $\beta = 0.0$ (independently-trained) | **boundary** | $47.2 \pm 14.1$ |
| | centre | $28.5 \pm 4.1$ |
| | ens. (mean) | $29.2 \pm 6.8$ |
| | intp. (max) | $38.2 \pm 7.8$ |
| CPS: 3 seen tasks of same coarse label, low-capacity | **boundary** | $49.4 \pm 10.8$ |
| | centre | $33.4 \pm 8.4$ |
| | ens. (mean) | $32.9 \pm 6.1$ |
| | intp. (max) | $53.4 \pm 8.5$ |

| Test Accuracy | CIFAR10 | 1 | 2 | 3 | 4 | 5 |
|---|---|---|---|---|---|---|
| Trained from scratch | 67.7 | 70.9 | 67.1 | 61.7 | 69.4 | 70.9 |
| Standard Hypernetwork (von Oswald et al., 2020) | | | | | | |
| Hypernetwork, evaluated after each task | 63.7 | 63.2 | 60.1 | 62.4 | 61.7 | 62.4 |
| Hypernetwork, evaluated after task 5 | 61.8 | 61.9 | 58.9 | 59.3 | 59.2 | 59.9 |
| CPS-regularized Hypernetwork | | | | | | |
| Hypernetwork, evaluated after each task | 65.6 | 68.2 | 64.4 | 68.7 | 64.8 | 64.9 |
| Hypernetwork, evaluated after task 5 | 63.6 | 65.0 | 60.7 | 62.9 | 61.6 | 61.7 |

Table 2: *CPS vs multiple test-time distributions:* Between $\beta = 0.0$ and $\beta = 1.0$, with respect to a set of $50^3$ interpolated tasks, we evaluate accuracy with respect to the lowest-loss boundary parameter, centre parameter, ensembling 1000 parameters and averaging their predictions, and searching for the lowest-loss interpolated parameter (excluding boundary parameters).

Table 3: *CPS vs multiple test-time distributions:* Test set accuracy in continual learnign setting on CIFAR10 and subsequent CIFAR100 splits of 10 classes. We evaluate von Oswald et al. (2020)'s hypernetwork and our CPS-regularized hypernetwork, and compare between training on each task individually, evaluating a task right after learning a new task, and evaluating a task after learning all tasks.

parameter space is higher than $\beta = 0.0$. In-line with Observation 1, we find from Appendix A.3.8 that, compared to a clean test-time distribution, we retain a comparable accuracy (centre / ensemble / interpolation) under different types of test-time perturbations.

To study improvements to task adaptation and meta learner capacity, we evaluate an additional CPS-regularization term to von Oswald et al. (2020)'s hypernetwork to find reduced catastrophic forgetting across tasks (Table 3). Though lowest-loss parameters are sparser in the compressed subspace, the hypernetwork can map parameters to tasks within the bounds of the parameter end-points. Though this is counter-intuitive as we would expect distant tasks to be located in distant subspaces in the input space, by forcing the parameters to be co-located in a single (yet interpolatable) subspace, we reduce the propensity for the hypernetwork to compute distant ($\beta = 0.0$) parameters with high-loss regions in between computed parameters.

**(Observation 4)** *Subspace capacity supports unseen tasks of different fine labels and different coarse labels.* From Appendix A.3.8, in addition to test set performance of seen tasks, we find that training a CPS on seen tasks of constant coarse label set can retain above-chance accuracy on unseen tasks of different fine labels as well as distinctly different coarse labels, supported by the high mean and low standard deviation. Semantic features may transfer across tasks of identical coarse labels, which can explain the performance with respect to unseen tasks of shared coarse labels. The performance with respect to unseen tasks of unshared coarse labels could be interpreted as the CPS containing large enough sets of (random) parameters that fit this task without training, in-line with Ramanujan

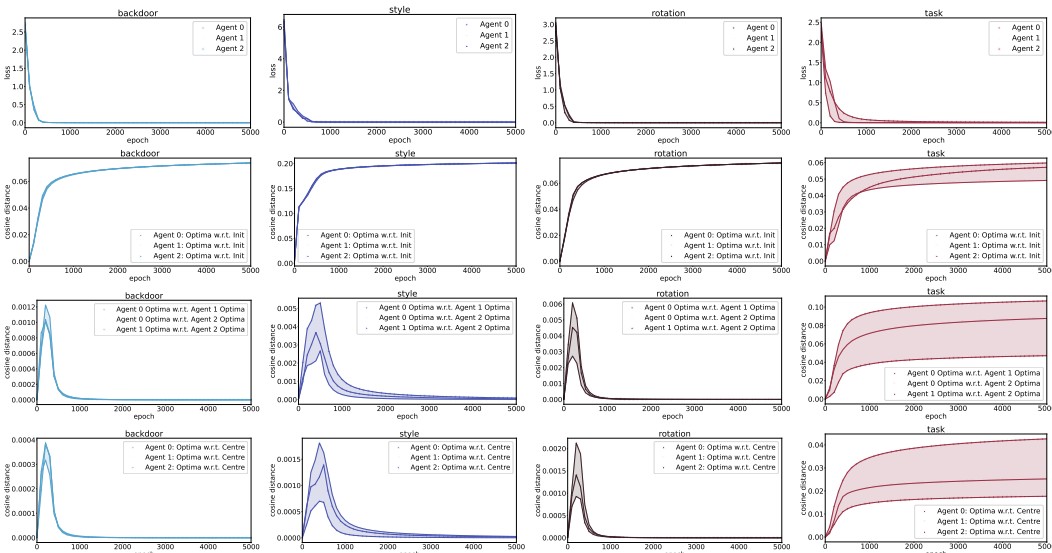

Figure 3: *Change in parameter subspace dynamics:* Loss and cosine distance per epoch during training of CPS with 3 train-time distributions ($\beta = 1.0$).

et al. (2019) where a randomly-initialized network contains low-loss subnetworks. These phenomena indicate an extremely large and flexible capacity supported within the parameter subspace; it has an increased number of low-loss parameters for shifted and unseen test-time distributions of shared labels, but also contains random parameters that can arbitrarily fit random tasks.

**(Observation 5)** *Sampled parameters can also be used as multi-task solutions.* To evaluate how well parameter point-estimates can operate as multi-task solutions, we consider over-parameterizaton of the subspace with respect to (i) minimizing the number of train-time distributions (which, according to Theorem 1, would reduce capacity per point-estimate allocated to an opposing end-point), and (ii) increasing model capacity (depth and/or width). Considering varying over-parameterization w.r.t. a single test-time distribution (Appendix A.3.4), though a larger capacity model can retain higher centre/ensemble/interpolation accuracy than a smaller capacity model, accuracy w.r.t. the test set of train-time distributions do not vary significantly when the number of train-time distributions varies (i.e. a trained subspace can expand or change capacity to support more parameter end-points accordingly). Similarly in Appendix A.3.7, when we vary the number of non-shared fine and coarse label sets, though the centre/ensemble accuracy does not improve significantly, we find that the accuracy of the lowest-loss interpolatable parameter increases with an increase in model capacity.

In Appendix A.3.8, unique-task solutions outperform multi-task solutions. From Appendices A.3.5 and A.3.6, we find that increasing model capacity and decreasing non-label-sharing tasks increases performance for multi-task learning: an over-parameterized subspace yields high accuracy when evaluated with the subspace centre or ensemble. An increase in centre/ensemble accuracy indicates that a sampled point has a higher propensity of providing low-loss inference on a sampled input set. As the number of non-label-shared tasks increases, the general performance decreases; increasing the capacity helps mitigate this, though it only increases the ability to locate a linearly-interpolated multi-task solution. When a subspace is over-parameterized for multi-task learning, though accuracy is increased for centre / ensemble / interpolation, it does not linearly minimize the gap between them.

## 6    CONCLUSION

We learn that sampling points in the parameter subspace can return lower-loss mapped parameters if the space was compressed during training, yield robust accuracy across various perturbation types, and reduces catastrophic forgetting when adapted into a hypernetwork. We motivate further study into enforcing geometric structure to parameter subspace construction and methods of test-time adaptation towards (joint) distribution shifts.

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

# A  APPENDIX

## A.1  SUPPORTING THEORY

**Definition 1.**  *A **distribution shift** is the divergence between a train-time distribution $\mathbf{x_0}, \mathbf{y_0}$ and a test-time distribution $\hat{\mathbf{x}}, \hat{\mathbf{y}}$, where $\hat{\mathbf{x}}, \hat{\mathbf{y}}$ is an interpolated distribution between $\mathbf{x_0}, \mathbf{y_0}$ and a target distribution $\mathbf{x_i^{\Delta}}, \mathbf{y_i^{\Delta}}$ such that $\hat{\mathbf{x}} = \sum_i^N \alpha_i \mathbf{x_i^{\Delta}}$ and $\hat{\mathbf{y}} = \mathbf{y_i}$ where $i = \arg\max_i \alpha_i$.*

Let $\mathcal{X}, \mathcal{Y}, \mathcal{K}, \Theta$ be denoted as the input space, coarse label space, fine label space, and parameter space respectively, such that $\mathcal{X} \mapsto \mathcal{Y}, \mathcal{X} \mapsto \mathcal{K}, \mathcal{K} \mapsto \mathcal{Y}, \mathcal{X} \mapsto \Theta$. Coarse labels are the super-class / higher-order labels of fine labels. $f$ is a base learner function that accepts inputs $\mathbf{x}, \theta$ to return predicted labels $\bar{\mathbf{y}} = f(\theta; \mathbf{x})$. The model parameters $\theta \sim \Theta$ are sampled from the parameter space, for example with an optimization procedure e.g. SGD, such that it minimizes the loss between the ground-truth and predicted labels: $\mathcal{L}(\theta; \mathbf{x}, \mathbf{y}) = \frac{1}{|\mathbf{x}|} \sum_i^{|\mathbf{x}|} (f(\theta; \mathbf{x}) - \mathbf{y})^2$.

In-line with notation in Pan & Yang (2010), for a marginal (or feature) distribution $\mathcal{P}(\mathcal{X})$ and a conditional (or label) distribution $\mathcal{P}(\mathcal{Y}|\mathcal{X})$, a joint distribution shift is denoted as a joint transformation in the marginal-conditional distributions. A covariate shift manifests as a change in the marginal distribution $\mathcal{P}(\mathcal{X})$ under constant conditional distribution $\mathcal{P}(\mathcal{Y}|\mathcal{X})$. A label shift manifests as a change in the conditional distribution $\mathcal{P}(\mathcal{Y}|\mathcal{X})$ under constant marginal distribution $\mathcal{P}(\mathcal{X})$. As elaborated in Section 2.1, deep neural networks may not perform well under this joint shift setting, and unifying robustness methods are limited. To simplify our language in discussion, we use a set of points sampled in a space to pertain to a distribution.

To retain generality in notation and implementation, we re-iterate the aforementioned formalism in terms of sampled points in their respective spaces. We sample mapped points in the input-label spaces as $\mathbf{x}, \mathbf{y} = \{\{x, y\} | \texttt{condition} \sim \mathcal{X}, \mathcal{Y}\}$. We describe shift here with respect to a single target label in order to highlight changes in the input-label mappings; the notation extends to non-singular label sets. Suppose we sample a clean / source / train-time set of points, constrained by a mapped target label $y^*$: $\mathbf{x_0}, \mathbf{y_0} = \{\{x, y\} | y = y^* \sim \mathcal{X}, \mathcal{Y}\}$.

A distribution shift is the divergence between a train-time distribution $\mathbf{x_0}, \mathbf{y_0}$ and a test-time distribution $\hat{\mathbf{x}}, \hat{\mathbf{y}}$, where $\hat{\mathbf{x}}, \hat{\mathbf{y}}$ is an interpolated distribution between $\mathbf{x_0}, \mathbf{y_0}$ and a target distribution $\mathbf{x_i^{\Delta}}, \mathbf{y_i^{\Delta}}$ such that $\hat{\mathbf{x}} = \sum_i^N \alpha_i \mathbf{x_i^{\Delta}}$ and $\hat{\mathbf{y}} = \mathbf{y_i} | i := \arg\max_i \alpha_i$. For a set of distributions $\{\mathbf{x_0} \mapsto \mathbf{y_0}, \mathbf{x_1^{\Delta}} \mapsto \mathbf{y_1^{\Delta}}, ..., \mathbf{x_N^{\Delta}} \mapsto \mathbf{y_N^{\Delta}}\}$ containing $(N-1)$ target distributions, we sample interpolation coefficients $\alpha_i \sim [0, 1]$ s.t. $\alpha_i \mapsto \mathbf{x_i}$ and $\sum_i^N \alpha_i \leq N$. $N$ is the number of distributions being used; in CPS training, it is the number of train-time distributions used in training the subspace; in task interpolation, it is the number of train/test-time distributions used in interpolating between task sets. A disjoint distribution shift is a distribution shift of only one target distribution $|\{\alpha_i\}| = 2$ such that $\hat{\mathbf{x}} = \alpha \mathbf{x^{\Delta}} + (1 - \alpha) \mathbf{x_0}$. A joint distribution shift is a distribution shift of multiple target distributions $|\{\alpha_i\}| > 2$ such that $\hat{\mathbf{x}} = \sum_i^N \alpha_i \mathbf{x_i^{\Delta}}$.

Each distinct target distribution is varied by the *type* of perturbation (e.g. label, domain, task) and the *variation* per type (e.g. multiple domains, multiple backdoor triggers). For a clean test-time distribution where $||\mathbf{x_1^{\Delta}} - \mathbf{x_0}||_p \to 0$ and $y = y^*$, no shift takes place. If $\hat{\mathbf{x}}, \hat{\mathbf{y}}$ manifests label shift ($y \neq y^*$, $\min ||\mathbf{x_1^{\Delta}} - \mathbf{x_0}||_p$), then $\mathbf{x_1^{\Delta}}, \mathbf{y_1^{\Delta}} = \{\{x, y\} | y \neq y^* \sim \mathcal{X}, \mathcal{Y}\}$. If $\hat{\mathbf{x}}, \hat{\mathbf{y}}$ manifests covariate shift ($||\mathbf{x_2^{\Delta}} - \mathbf{x_0}||_p > 0$), such as domain shift, then $\mathbf{x_2^{\Delta}}, \mathbf{y_2^{\Delta}} = \{\{x, y\} | y = y_{\text{ambg}} \sim \mathcal{X}, \mathcal{Y}\}$, where the sampled $x$ may not correspond to a clearly-defined class (i.e. a random / ambiguous class $y_{\text{ambg}}$). If $\hat{\mathbf{x}}, \hat{\mathbf{y}}$ manifests task shift (unrestricted $||\mathbf{x_3^{\Delta}} - \mathbf{x_0}||_p > 0$); for label-shared task shift (identical coarse labels $y$, non-identical fine labels $k$) $\mathbf{x_3^{\Delta}}, \mathbf{y_3^{\Delta}} = \{\{x, y\} | y = y^*, k \neq k^* \sim \mathcal{X}, \mathcal{Y}, \mathcal{K}\}$; for non-label-shared task shift (non-identical coarse nor fine labels) $\mathbf{x_3^{\Delta}}, \mathbf{y_3^{\Delta}} = \{\{x, y\} | y \neq y^*, k \neq k^* \sim \mathcal{X}, \mathcal{Y}, \mathcal{K}\}$.

For interpolated input sets between $\mathbf{x_0}$ and $\mathbf{x_i^{\Delta}}$, we would expect $0 < \alpha_0, \alpha_i < 1$, unless we expect the shifted distribution to completely manifest the target distribution (e.g. rotated sets, end-point task sets) such that $\alpha_i = 1$. The labels of the interpolated set is the label set with the larger corresponding $\alpha_i$. In label shift, the perturbations are small enough that we retain the ground-truth labels. For label-shared task shift, the ground-truth (coarse) labels between the end-point tasks are identical. For non-label-shared task shift, we do not interpolate between non-identical label sets between the end-point tasks.

If we randomly-interpolate random input sets with random or ambiguous label assignments, we incur the risk of (i) the perturbed input set as being as being perceptually inconsistent with the assigned label, and (ii) the model overfitting towards random label assignments (Zhang et al., 2017a). To mitigate these concerns and generate semantically-valid (perceptually-interpreted as belonging to its assigned label) perturbations, we adopt existing perturbation generation methods that have also been human-evaluated as perceptually consistent. The use of these methods can be interpreted as the filtering of invalid points in the input space for the evaluation of mapping between $\mathcal{X}$ and $\Theta$.

**Corollary 1.** *Test-time distributions can be decomposed into a weighted sum between known train-time distributions and computable interpolation coefficients.*

We assume a given test-time distribution $\hat{\mathbf{x}}$ can be decomposed into a sum of parts $\hat{\mathbf{x}} := \sum_i^N \alpha_i \mathbf{x_i}$, where $N$ is the number of component distributions. For the component distributions $\{\mathbf{x_i}\}^N$, if a subset of the distributions exist in the set of train-time distributions, then the interpolation coefficient value per train-time distribution lies in $0 < \alpha_i \leq 1$; if none of the distributions exist in the set of train-time distributions, then the interpolation coefficient value per train-time distribution is 0.

**Corollary 2.** *For an input interpolated between a set of input end-points with interpolation coefficients $\{\alpha_i\}^N$, the loss term per epoch can be decomposed into component/end-point loss terms of equivalent interpolation coefficients $\{\alpha_i\}^N$, but the parameters may or may not be decomposed into component/end-point parameters of $\{\alpha_i\}^N$.*

The loss of an interpolated set with respect to a parameter updated till epoch $e$ can be computed with respect to the weighted sum of the loss with respect to the boundary input points and interpolation coefficients for the interpolated input set.

$$
\begin{aligned}
\frac{\partial \mathcal{L}(\theta_{t=e}; \hat{\mathbf{x}})}{\partial \theta_{t=e}} &= \frac{(\theta_{t=e} \sum_i^N \alpha_i \mathbf{x_i} - \mathbf{y}) - (\theta_{t=e-1} \sum_i^N \alpha_i \mathbf{x_i} - \mathbf{y})}{\theta_{t=e} - \theta_{t=e-1}} \\
&= \sum_i^N \frac{(\theta_{t=e} \alpha_i \mathbf{x_i} - \mathbf{y}) - (\theta_{t=e-1} \alpha_i \mathbf{x_i} - \mathbf{y})}{\theta_{t=e} - \theta_{t=e-1}} \\
&= \sum_i^N \alpha_i \frac{\partial \mathcal{L}(\theta_{t=e}; \mathbf{x_i})}{\partial \theta_{t=e}}
\end{aligned}
\tag{2}
$$

$\theta_{t=e}$ is computed with respect to the total loss at epoch $e-1$. $\theta_{t=e-1}$ updated with respect to multiple $\mathbf{x_i}$ may not be identical to $\theta_{t=e-1}$ updated with respect to a single set $\mathbf{x_0}$. As a result, we cannot claim that an interpolated input set must correspond to or be satisfied with a parameter composed of the weighted sum of the interpolation coefficients and end-point parameters $\theta^* \neq \sum_i^N \alpha_i \theta_i$.

**Corollary 3.** *Training and inference with a single parameter point-estimate $\theta$ may face a trade-off in loss w.r.t. $X_i$ and $X_n$ if interference exists between $X_i$ and $X_n$.*

At a given epoch $e$, if $\frac{\partial \mathcal{L}(\theta; X_i)}{\partial \theta} \cdot \frac{\partial \mathcal{L}(\theta; X_n)}{\partial \theta} < 0$, then the decrease in loss w.r.t. one subset (either $X_i$ or $X_n$) would result in the increase in loss of the other during training, and consequently the same during inference.

**Assumptions/Constraints.** We also note a few (non-exhaustive) assumptions or criteria for interpreting the problem and evaluating the CPS.

1. *Each parameter point-estimate in the parameter space $\Theta$ can be evaluated against more than one input distribution.*

2. *The input space and parameter space are continuous, and interpolated points can be sampled from both spaces.*

3. *A low-loss parameter point-estimate for a test-time distribution $\theta \mapsto \hat{\mathbf{x}}$ should be computable with minimal storage and time complexity.*

4. *We assume no task index or conditioning input or meta-data of the test-time distribution.*

**Theorem 1.** *For $N + 1$ parameters trained in parallel, any individual parameter $\theta_i$ is a function of its mapped data $X_i \mapsto Y_i$ as well as the remaining parameters $\{\theta_n\}^{n \in N}$ (and by extension, $X_n \mapsto Y_n$). Hence, a linearly-interpolated parameter between $\theta_i$ and $\{\theta_n\}^{n \in N}$ is a function of $\{\theta_n\}^{n \in N}$ and $\theta_i | \{\theta_n\}^{n \in N}$.*

*Proof.* Suppose $N + 1$ parameters are trained in parallel, and the number of dimensions of each parameter is $M = |\theta|$. To simplify the analysis, we use the Euclidean norm distance, and make relevant approximations.

First, we decompose the loss w.r.t. $\theta_i$, in order to obtain $\theta_i$ in terms of $X_i$, $Y_i$, and $\{\theta_n\}^{n \in N}$ such that $\mathcal{L}(\theta_i) = 0$. We denote $\mathcal{C} = \sum_n^N \theta_n$. By the Cauchy-Schwarz ineuality, we apply the following term replacement: $(\sum_n^N \theta_n)^2 = (\sum_n^N 1 \cdot \theta_n)^2 \leq N \cdot \sum_n^N \theta_n^2 \Rightarrow \frac{\mathcal{C}}{N} \leq \sum_n^N \theta_n^2$.

$$
\begin{aligned}
\mathcal{L}(\theta_i) &= \mathcal{L}(\theta_i, X_i, Y_i) + \texttt{dist}(\theta_i, \{\theta_n\}^{n \in N}) \\
&= (\theta_i X_i - Y_i) + \sum_n^N \sum_m^M (\theta_{i,m} - \theta_{n,m})^2 \\
&\approx (\theta_i X_i - Y_i) + \sum_n^N (\theta_i^2 - 2\theta_i \theta_n + \theta_n^2) \\
&\approx (\theta_i X_i - Y_i) + N\theta_i^2 - 2\theta_i \mathcal{C} + \frac{\mathcal{C}^2}{N} \\
&\approx N\theta_i^2 + (X_i - 2\mathcal{C})\theta_i + (\frac{\mathcal{C}^2}{N} - Y_i) \\
\theta_i &:= \frac{2\mathcal{C} - X_i \pm \sqrt{(X_i - 2\mathcal{C})^2 - 4N(\frac{\mathcal{C}^2}{N} - Y_i)}}{2N} \quad \text{s.t.} \quad \mathcal{L}(\theta_i) = 0 \\
\theta_i &:= \frac{2\mathcal{C} - X_i \pm \sqrt{-4X_i \mathcal{C} + 4NY_i + X_i^2}}{2N}
\end{aligned}
\tag{3}
$$

Next, we compute the change in $\theta_i$ w.r.t. change in $\mathcal{C}$, and find that the $\mathcal{C}$ term is persistent (non-eliminated) and is retained in the gradient calculation and optimization procedure of $\theta_i$.

$$
\begin{aligned}
\frac{\partial \theta_i}{\partial \mathcal{C}} &= \frac{1}{2N} \left[ 2 \pm \frac{-4X_i}{2\sqrt{-4X_i \mathcal{C} + 4NY_i + X_i^2}} \right] \\
&= \frac{1}{N} \pm \frac{-2X_i}{N\sqrt{-4X_i \mathcal{C} + 4NY_i + X_i^2}}
\end{aligned}
\tag{4}
$$

In order for Eqt 4 to remain valid, Eqt 5 needs to be satisfied, which also shows that $\mathcal{C}$ is dependent or contains artifacts from $X_i \mapsto Y_i$.

$$
\begin{aligned}
-4X_i \mathcal{C} + 4NY_i + X_i^2 &\geq 0 \\
\mathcal{C} &\leq \frac{4X_i}{4NY_i + X_i^2}
\end{aligned}
\tag{5}
$$

Further, suppose we linearly-interpolate a parameter $\theta$ within the bounded space of $\theta_i \cup \{\theta_n\}^{n \in N}$, and the weighting per parameter is $\alpha_n \mapsto \theta_n$ such that $\sum_n^{N+1} \alpha_n \leq N + 1$. We find that the interpolated parameter $\theta$ has a non-eliminated term $\mathcal{C}$ regardless of set $\alpha$ or position in the bounded parameter space.

$$
\begin{aligned}
\theta &= \alpha_i \theta_i + \sum_{n, n \neq i}^N \alpha_n \theta_n \\
\frac{\partial \theta}{\partial \mathcal{C}} &\approx \alpha_i \frac{\partial \theta_i}{\partial \mathcal{C}} + \frac{1}{N} \sum_{n, n \neq i}^N \alpha_n
\end{aligned}
\tag{6}
$$

$\square$

**Theorem 2.** *For $N + 1$ parameters trained in parallel, for an individual parameter $\theta_i$ and another parameter $\theta_n$ where $n \in N$, the change in loss per iteration of SGD with respect to each parameter $\frac{\partial \mathcal{L}(\theta_i; X_i)}{\partial \theta_i}$, $\frac{\partial \mathcal{L}(\theta_n; X_n)}{\partial \theta_n}$ will both be in the same direction (i.e. identical signs), and will tend towards a local region of the parameter space upon convergence of loss.*

*Proof.* Suppose $N + 1$ parameters are trained in parallel, and the number of dimensions of each parameter is $M = |\theta|$. We evaluate the incremental loss per iteration $\forall e \in E$, which is the distance regularization term.

$$
\begin{aligned}
\Delta\mathcal{L}(\theta_i)_e &= \sum_n^N \sum_m^M (\theta_{i,m} - \theta_{n,m})^2 \\
&\approx \sum_n^N (\theta_i - \theta_n)^2 \\
&:= \sum_n^N \left[ \left( \theta_i - \frac{\partial \mathcal{L}(\theta_i; X_i)}{\partial \theta_i} \right) - \left( \theta_n - \frac{\partial \mathcal{L}(\theta_n; X_n)}{\partial \theta_n} \right) \right]^2 \\
&= \sum_n^N \left[ (\theta_i - \theta_n) + \left( \frac{\partial \mathcal{L}(\theta_n; X_n)}{\partial \theta_n} - \frac{\partial \mathcal{L}(\theta_i; X_i)}{\partial \theta_i} \right) \right]^2 \\
&= \sum_n^N \left[ (\theta_i - \theta_n)^2 + 2(\theta_i - \theta_n)\left( \frac{\partial \mathcal{L}(\theta_n; X_n)}{\partial \theta_n} - \frac{\partial \mathcal{L}(\theta_i; X_i)}{\partial \theta_i} \right) + \left( \frac{\partial \mathcal{L}(\theta_n; X_n)}{\partial \theta_n} - \frac{\partial \mathcal{L}(\theta_i; X_i)}{\partial \theta_i} \right)^2 \right]
\end{aligned}
\tag{7}
$$

Given that minimizing the last term can result in minimizing the loss $\min \Delta\mathcal{L}(\theta_i)_e := \min(\theta_n - \frac{\partial \mathcal{L}(\theta_n; X_n)}{\partial \theta_n} - \frac{\partial \mathcal{L}(\theta_i; X_i)}{\partial \theta_i})$, we next show that the gradient update (change in loss per iteration of SGD with respect to each parameter) terms at iteration $e$ must be positive and share the same direction when loss converges to 0. For $\Delta\mathcal{L}(\theta_i)_e \geq 0$ and $\forall n$ where $n \in N$,

$$
\begin{aligned}
\left( \frac{\partial \mathcal{L}(\theta_n; X_n)}{\partial \theta_n} \right)^2 + \left( \frac{\partial \mathcal{L}(\theta_i; X_i)}{\partial \theta_i} \right)^2 - 2\left( \frac{\partial \mathcal{L}(\theta_i; X_i)}{\partial \theta_i} \right)\left( \frac{\partial \mathcal{L}(\theta_n; X_n)}{\partial \theta_n} \right) &\geq 0 \\
\frac{\partial \mathcal{L}(\theta_i; X_i)}{\partial \theta_i} \cdot \frac{\partial \mathcal{L}(\theta_n; X_n)}{\partial \theta_n} &\leq \frac{1}{2}\left[ \left( \frac{\partial \mathcal{L}(\theta_n; X_n)}{\partial \theta_n} \right)^2 + \left( \frac{\partial \mathcal{L}(\theta_i; X_i)}{\partial \theta_i} \right)^2 \right]
\end{aligned}
\tag{8}
$$

For $\Delta\mathcal{L}(\theta_i)_e \equiv 0$,

$$
\frac{\partial \mathcal{L}(\theta_i; X_i)}{\partial \theta_i} \cdot \frac{\partial \mathcal{L}(\theta_n; X_n)}{\partial \theta_n} \equiv \frac{1}{2}\left[ \left( \frac{\partial \mathcal{L}(\theta_n; X_n)}{\partial \theta_n} \right)^2 + \left( \frac{\partial \mathcal{L}(\theta_i; X_i)}{\partial \theta_i} \right)^2 \right] \geq 0
\tag{9}
$$

Hence, $\frac{\partial \mathcal{L}(\theta_i; X_i)}{\partial \theta_i} \cdot \frac{\partial \mathcal{L}(\theta_n; X_n)}{\partial \theta_n}$ must be positive, and the component terms must be in the same direction (identical signs).

To further simplify analysis, we evaluate loss updates from iteration $e = 0$ to $e = E - 1$ where loss converges at iteration $E$, thus the component loss terms are non-zero. The two loss terms are thus non-orthogonal, co-directional, and the angle between them is less than $90°$.

$$
\frac{\partial \mathcal{L}(\theta_i; X_i)}{\partial \theta_i} \cdot \frac{\partial \mathcal{L}(\theta_n; X_n)}{\partial \theta_n} > 0
\tag{10}
$$

Given that $\theta_i$ and $\theta_n$ are initialized at the same constant random initialization $\theta_{i,e=0} \equiv \theta_{n,e=0}$, the change in each $\theta$ from iterations $e = 0$ to $e = E - 1$ is attributed to the difference in the summation of the gradient updates.

$$\begin{cases} \theta_i := \theta_{i,t=0} - \sum_{e}^{E-1} \dfrac{\partial \mathcal{L}(\theta_{i,t=e}; X_i)}{\partial \theta_{i,t=e}} & (11) \\[3em] \theta_n := \theta_{n,t=0} - \sum_{e}^{E-1} \dfrac{\partial \mathcal{L}(\theta_{n,t=e}; X_n)}{\partial \theta_{n,t=e}} & (12) \end{cases}$$

We show next, that through enforcing distance regularization, we can exert a tendency for the distance between the summation of the loss term components to be smaller if they share the same direction, than if they were orthogonal or opposing directions.

$$\left| \left[ \left| \sum_{e}^{E-1} \frac{\partial \mathcal{L}(\theta_{i,t=e}; X_i)}{\partial \theta_{i,t=e}} \right| - \left| \sum_{e}^{E-1} \frac{\partial \mathcal{L}(\theta_{n,t=e}; X_n)}{\partial \theta_{n,t=e}} \right| \right] \Big|_{\frac{\partial \mathcal{L}(\theta_i; X_i)}{\partial \theta_i} \cdot \frac{\partial \mathcal{L}(\theta_n; X_n)}{\partial \theta_n} > 0} \right|$$
$$< \left| \left[ \left| \sum_{e}^{E-1} \frac{\partial \mathcal{L}(\theta_{i,t=e}; X_i)}{\partial \theta_{i,t=e}} \right| - \left| \sum_{e}^{E-1} \frac{\partial \mathcal{L}(\theta_{n,t=e}; X_n)}{\partial \theta_{n,t=e}} \right| \right] \Big|_{\frac{\partial \mathcal{L}(\theta_i; X_i)}{\partial \theta_i} \cdot \frac{\partial \mathcal{L}(\theta_n; X_n)}{\partial \theta_n} \leq 0} \right| \tag{13}$$

Thus we show that $\theta_i$ and $\theta_n$ are closer in distance in the parameter space. This can result in $\theta_i$ and $\theta_n$ sharing sub-matrices (intrinsic dimensions), and they may reside in the same or neighboring subspace.

$\square$

**Theorem 3.** *For $N+1$ parameters trained in parallel, the expected distance between the optimal parameter of an interpolated input $\hat{\mathbf{x}} \mapsto \hat{\theta}$ would be smaller with respect to sampled points in a distance-regularized subspace than non-distance-regularized subspace.*

Informally, we conclude from Theorems 1 and 2 that, though non-linear and/or linear point-estimates may exist in a non-compressed parameter subspace, a compressed parameter subspace contains less interference attributed to changes in other dimensions; in other words, the CPS method reduces the dimensions of interpolation. Practically, this can manifest as most values in the weights matrix being relatively frozen or constant (and only few values or a submatrix being changed), or the range per value in a weights matrix may be narrower.

*Proof.* Suppose we sample an interpolated input $\hat{\mathbf{x}} = \alpha_i \mathbf{x_i} + \alpha_n \mathbf{x_n}$. All trained parameters begin with a constant and equal initialization at $t = 0$.

We would like to show here that any randomly interpolated input set of coefficients $\alpha_i$, $\alpha_n$ would be closer in distance to any parameter point-estimate interpolated between CPS-regularized parameter end-points with coefficients $\gamma_i$, $\gamma_n$, compared to non-CPS-regularized.

First we state the loss terms for a ground-truth parameter $\hat{\theta}$ corresponding to $\hat{\mathbf{x}}$. Recall from Theorem 2 that the loss terms in the end-point parameters converge in the same direction $\frac{\partial \mathcal{L}(\theta_i; X_i)}{\partial \theta_i} \cdot \frac{\partial \mathcal{L}(\theta_n; X_n)}{\partial \theta_n} > 0$. We find here a similarity to CPS, where in order for the loss to converge $\mathcal{L}(\hat{\theta}) \to 0$ the product of the loss terms for the ground-truth parameter also converges in the same direction $\frac{\partial \mathcal{L}(\hat{\theta}; X_i)}{\partial \hat{\theta}} \cdot \frac{\partial \mathcal{L}(\hat{\theta}; X_n)}{\partial \hat{\theta}} > 0$.

$$\hat{\theta} := \hat{\theta} - \sum_{e}^{E-1} \left[ \frac{\partial \mathcal{L}(\hat{\theta}_{t=e}; \alpha_i \mathbf{x_i} + \alpha_n \mathbf{x_n})}{\partial \hat{\theta}_{t=e}} \right]$$
$$= \hat{\theta} - \sum_{e}^{E-1} \left[ \alpha_i \frac{\partial \mathcal{L}(\hat{\theta}_{t=e}; \mathbf{x_i})}{\partial \hat{\theta}_{t=e}} + \alpha_n \frac{\partial \mathcal{L}(\hat{\theta}_{t=e}; \mathbf{x_n})}{\partial \hat{\theta}_{t=e}} \right] \quad \text{(by Corollary 2)} \tag{14}$$

Next, as the summation of loss per epoch indicates the region in the parameter space with respect to a constant initialization (as in Theorem 2), we compute the distance in the summation of loss for $\hat{\theta}$ compared to the boundary or end-point parameters of an uncompressed space. As the component

loss terms are not regularized with respect to each other, their product may or may not be greater than 0, i.e. their product is in the range [-1, 1].

$$
\begin{aligned}
\mathbb{E}\Bigg[ & \Bigg| \Bigg[ \alpha_i \frac{\partial \mathcal{L}(\hat{\theta}_{t=e}; \mathbf{x_i})}{\partial \hat{\theta}_{t=e}} + \alpha_n \frac{\partial \mathcal{L}(\hat{\theta}_{t=e}; \mathbf{x_n})}{\partial \hat{\theta}_{t=e}} - \gamma_i \frac{\partial \mathcal{L}(\theta_{i,t=e}; \mathbf{x_i})}{\partial \theta_{i,t=e}} \Bigg|_{\frac{\partial \mathcal{L}(\theta_i; \mathbf{x_i})}{\partial \theta_i} \cdot \frac{\partial \mathcal{L}(\theta_n; \mathbf{x_n})}{\partial \theta_n} \sim [-1,1]} \Bigg] \\
& - \Bigg[ \alpha_i \frac{\partial \mathcal{L}(\hat{\theta}_{t=e}; \mathbf{x_i})}{\partial \hat{\theta}_{t=e}} + \alpha_n \frac{\partial \mathcal{L}(\hat{\theta}_{t=e}; \mathbf{x_n})}{\partial \hat{\theta}_{t=e}} - \gamma_n \frac{\partial \mathcal{L}(\theta_{n,t=e}; \mathbf{x_n})}{\partial \theta_{n,t=e}} \Bigg|_{\frac{\partial \mathcal{L}(\theta_n; \mathbf{x_i})}{\partial \theta_i} \cdot \frac{\partial \mathcal{L}(\theta_n; \mathbf{x_n})}{\partial \theta_n} \sim [-1,1]} \Bigg] \Bigg| \Bigg] \\
= \mathbb{E}\Bigg[ & \Bigg| 2\Bigg[ \alpha_i \frac{\partial \mathcal{L}(\hat{\theta}_{t=e}; \mathbf{x_i})}{\partial \hat{\theta}_{t=e}} + \alpha_n \frac{\partial \mathcal{L}(\hat{\theta}_{t=e}; \mathbf{x_n})}{\partial \hat{\theta}_{t=e}} \Bigg] \\
& + \Bigg[ \gamma_n \frac{\partial \mathcal{L}(\theta_{n,t=e}; \mathbf{x_n})}{\partial \theta_{n,t=e}} \Bigg|_{\frac{\partial \mathcal{L}(\theta_n; \mathbf{x_i})}{\partial \theta_i} \cdot \frac{\partial \mathcal{L}(\theta_n; \mathbf{x_n})}{\partial \theta_n} \sim [-1,1]} - \gamma_i \frac{\partial \mathcal{L}(\theta_{i,t=e}; \mathbf{x_i})}{\partial \theta_{i,t=e}} \Bigg|_{\frac{\partial \mathcal{L}(\theta_n; \mathbf{x_i})}{\partial \theta_i} \cdot \frac{\partial \mathcal{L}(\theta_n; \mathbf{x_n})}{\partial \theta_n} \sim [-1,1]} \Bigg] \Bigg| \Bigg]
\end{aligned}
\tag{15}
$$

Then, we evaluate the summation of loss for $\hat{\theta}$ against the boundary or end-point parameters of a compressed space.

$$
\begin{aligned}
\mathbb{E}\Bigg[ & \Bigg| \Bigg[ \alpha_i \frac{\partial \mathcal{L}(\hat{\theta}_{t=e}; \mathbf{x_i})}{\partial \hat{\theta}_{t=e}} + \alpha_n \frac{\partial \mathcal{L}(\hat{\theta}_{t=e}; \mathbf{x_n})}{\partial \hat{\theta}_{t=e}} - \gamma_i \frac{\partial \mathcal{L}(\theta_{i,t=e}; \mathbf{x_i})}{\partial \theta_{i,t=e}} \Bigg|_{\frac{\partial \mathcal{L}(\theta_i; \mathbf{x_i})}{\partial \theta_i} \cdot \frac{\partial \mathcal{L}(\theta_n; \mathbf{x_n})}{\partial \theta_n} > 0} \Bigg] \\
& - \Bigg[ \alpha_i \frac{\partial \mathcal{L}(\hat{\theta}_{t=e}; \mathbf{x_i})}{\partial \hat{\theta}_{t=e}} + \alpha_n \frac{\partial \mathcal{L}(\hat{\theta}_{t=e}; \mathbf{x_n})}{\partial \hat{\theta}_{t=e}} - \gamma_n \frac{\partial \mathcal{L}(\theta_{n,t=e}; \mathbf{x_n})}{\partial \theta_{n,t=e}} \Bigg|_{\frac{\partial \mathcal{L}(\theta_n; \mathbf{x_i})}{\partial \theta_i} \cdot \frac{\partial \mathcal{L}(\theta_n; \mathbf{x_n})}{\partial \theta_n} > 0} \Bigg] \Bigg| \Bigg] \\
= \mathbb{E}\Bigg[ & \Bigg| 2\Bigg[ \alpha_i \frac{\partial \mathcal{L}(\hat{\theta}_{t=e}; \mathbf{x_i})}{\partial \hat{\theta}_{t=e}} + \alpha_n \frac{\partial \mathcal{L}(\hat{\theta}_{t=e}; \mathbf{x_n})}{\partial \hat{\theta}_{t=e}} \Bigg] \\
& + \Bigg[ \gamma_n \frac{\partial \mathcal{L}(\theta_{n,t=e}; \mathbf{x_n})}{\partial \theta_{n,t=e}} \Bigg|_{\frac{\partial \mathcal{L}(\theta_n; \mathbf{x_i})}{\partial \theta_i} \cdot \frac{\partial \mathcal{L}(\theta_n; \mathbf{x_n})}{\partial \theta_n} > 0} - \gamma_i \frac{\partial \mathcal{L}(\theta_{i,t=e}; \mathbf{x_i})}{\partial \theta_{i,t=e}} \Bigg|_{\frac{\partial \mathcal{L}(\theta_n; \mathbf{x_i})}{\partial \theta_i} \cdot \frac{\partial \mathcal{L}(\theta_n; \mathbf{x_n})}{\partial \theta_n} > 0} \Bigg] \Bigg| \Bigg] \\
< \mathbb{E}\Bigg[ & \Bigg| 2\Bigg[ \alpha_i \frac{\partial \mathcal{L}(\hat{\theta}_{t=e}; \mathbf{x_i})}{\partial \hat{\theta}_{t=e}} + \alpha_n \frac{\partial \mathcal{L}(\hat{\theta}_{t=e}; \mathbf{x_n})}{\partial \hat{\theta}_{t=e}} \Bigg] && \text{(by Theorem 2)} \\
& + \Bigg[ \gamma_n \frac{\partial \mathcal{L}(\theta_{n,t=e}; \mathbf{x_n})}{\partial \theta_{n,t=e}} \Bigg|_{\frac{\partial \mathcal{L}(\theta_n; \mathbf{x_i})}{\partial \theta_i} \cdot \frac{\partial \mathcal{L}(\theta_n; \mathbf{x_n})}{\partial \theta_n} \sim [-1,1]} - \gamma_i \frac{\partial \mathcal{L}(\theta_{i,t=e}; \mathbf{x_i})}{\partial \theta_{i,t=e}} \Bigg|_{\frac{\partial \mathcal{L}(\theta_n; \mathbf{x_i})}{\partial \theta_i} \cdot \frac{\partial \mathcal{L}(\theta_n; \mathbf{x_n})}{\partial \theta_n} \sim [-1,1]} \Bigg] \Bigg| \Bigg]
\end{aligned}
\tag{16}
$$

We find that a ground-truth interpolated parameter is closer to a compressed space than an uncompressed space. This motivates our work that a compressed subspace contains a higher density of low-loss interpolated parameters than an uncompressed subspace.

$\square$

## A.2 EXPERIMENT CONFIGURATIONS

### A.2.1 ADAPTATION METHODS

**Data Augmentation** (e.g., CutMix (Yun et al., 2019a) or MixUp (Zhang et al., 2018)) is a common method to robustify models against domain shift (Huang et al., 2018), adversarial attacks (Zeng et al., 2020), and backdoor attacks (Borgnia et al., 2020). We implement CutMix (Yun et al., 2019a), where augmentation takes place per batch, and training completes in accordance with aforementioned early stopping. In the case of CutMix, for example, instead of removing pixels and filling them with black or grey pixels or Gaussian noise, we replace the removed regions with a patch from another image, while the ground truth labels are mixed proportionally to the number of pixels of combined images. 50% of the defender's allocation of the dataset is assigned to augmentation.

**Adversarial Training** is a method that increases model robustness by injecting adversarial examples into the training set, and commonly used against adversarial attacks (Goodfellow et al., 2015), backdoor attacks (Geiping et al., 2021), and domain adaptation (Ganin et al., 2016a). We retain the same adversarial configurations as in the adversarial attack (PGD, $\varepsilon = 0.4$). At each epoch, $50\%$ of the defender's allocation of the dataset is adversarially-perturbed.

**Backdoor Adversarial Training** (Geiping et al., 2021) extend the concept of adversarial training on defender-generated backdoor examples to insert their own triggers to existing labels. We implement backdoor adversarial training (Geiping et al., 2021), where the generation of backdoor perturbations is through Random-BadNet (Datta & Shadbolt, 2022)), where $50\%$ of the defender's allocation of the dataset is assigned to backdoor perturbation, $p, \varepsilon = 0.4$, and 20 different backdoor triggers used (i.e. allocation of defender's dataset for each backdoor trigger pattern is $(1 - 0.5) \times 0.8 \times \frac{1}{20}$ ).

**Fast Geometric Ensembles (Garipov et al., 2018)** train multiple networks to find their respective modes such that a high-accuracy path is formed that connects the modes in the parameter space. The method collects parameters at different checkpoints, which can subsequently be used as an ensemble. We retain the same model configurations as the clean setting (3-layer CNN), train 5 ensembles on randomly-sampled random initializations until the early-stopping condition.

**CPS: Sampling**

- Inferencing w.r.t. the **centre** of the subspace is denoted as computing a prediction $\bar{y} = f(\theta^*; \hat{\mathbf{x}})$ with respect to a given input and the centre parameter point-estimate $\theta^* = \sum_i^N \frac{1}{N} \theta_i$ (the average of the end-point parameters). This form of inference has also been used in Wortsman et al. (2021).

- Inferencing w.r.t. an **ensemble** in the subspace is denoted as computing a mean prediction $\bar{y} = \frac{1}{M} \sum_j^M f(\theta_j^*; \hat{\mathbf{x}})$ with respect to a given input and $M$ randomly-sampled parameter point-estimates in the subspace (we randomly sample $M = 1000$ interpolation coefficients $\{\alpha_{j,i}\}^{M \times N}$ to return an ensemble set of parameter point estimates $\{\theta_j^*\}^M = \{\alpha_{j,i}\}^{M \times N} \cdot \{\theta_i\}^N$. For our ensemble, we evaluate against the mean prediction. This form of inference has also been used in Wortsman et al. (2021); Garipov et al. (2018); Fort et al. (2020).

- Inferencing w.r.t. the maximum-accuracy (lowest-loss) **interpolated** point-estimate in the subspace is denoted as computing a prediction $\bar{y} = f(\theta_{j^*}; \hat{\mathbf{x}})$ with respect to a given input and the lowest-loss parameter point-estimate where $j^* := \arg\min_{j \sim M} \mathcal{L}(\theta_j; \mathbf{x}, \mathbf{y})$; this is specifically a unique-task solution, where a low-loss parameter maps back to one task/distribution $\hat{\mathbf{x}} \mapsto \theta_{j^*}$. For a multi-task solution, a low-loss parameter is mapped back to a set of $T$ task/distributions $\{\hat{\mathbf{x}}_{\mathbf{t}}\}^T \mapsto \theta_{j^*}$, where $j^* := \arg\min_{j \sim M} \sum_t^T \mathcal{L}(\theta_j; \mathbf{x_t}, \mathbf{y_t})$. We linearly-space 50 segments per parameter end-point pair (resulting in $50^3$ point-estimates to evaluate). There is enough distance between parameter end-points such that the interpolated points are not close approximations of the end-point (further verifiable from Figure 2). Inferencing w.r.t. interpolated points has also been used in Neyshabur et al. (2020).

- An additional baseline used in Table 2 is inferencing w.r.t. the lowest-loss **boundary** parameter in the subspace, denoted as computing a prediction $\bar{y} = f(\theta_{i^*}; \hat{\mathbf{x}})$ with respect to a given input and lowest-loss boundary parameter, where the latter is the parameter from a set of $N$ boundary or end-point parameters that returns the lowest loss $i^* := \arg\min_{i \sim N} \mathcal{L}(\theta_i; \mathbf{x}, \mathbf{y})$. This baseline informs us whether the lowest-loss interpolated parameter can outperform an uninterpolated SGD-trained parameter. In Table 2 specifically, we do not include the boundary parameters in the interpolated parameters set (interpolation coefficients: [0,0,1],[0,1,0],[1,0,0]), so as to ensure the 2 sets are mutually-exclusive and avoid the reuse of the boundary parameter as the lowest-loss interpolated parameter; for example, in case there is another point that can also achieve similar accuracy.

**CPS: Hypernetworks** A hypernetwork $h(\mathbf{x}, I) = f(\mathbf{x}; mf(\theta_{mf}; I))$ is a pair of learners, the base learner $f : \mathcal{X} \mapsto \mathcal{Y}$ and meta learner $mf : \mathcal{I} \mapsto \Theta_f$, such that for the conditioning input $I$ of input $\mathbf{x}$ (where $\mathcal{X} \mapsto \mathcal{I}$), $mf$ produces the base learner parameters $\theta_I = mf(\theta_{mf}; I)$. The function $mf(\theta_{mf}; I)$ takes a conditioning input $I$ to returns parameters $\theta_I \in \Theta_f$ for $f$. The meta learner parameters and base learner (of each respective distribution) parameters reside in their distinct parameter spaces $\theta_{mf} \in \Theta_{mf}$ and $\theta_f \in \Theta_f$. The learner $f$ takes an input $\mathbf{x}$ and returns an output $\bar{y} = f(\theta_I; \mathbf{x})$ that depends on both $\mathbf{x}$ and the task-specific input $I$. $T$ is number of tasks, $t$ is index of specific task being evaluated ($t^*$ being current task), $\omega$ is task regularizer (from von Oswald et al. (2020) implementation). We retain all other implementation configurations, such as the use of Adam

optimizer, setting task regularizer $\omega = 0.01$. $\theta_{mf}$ is the hypernetwork parameters at the current task's training timestep, $\theta_{mf}^*$ is the hypernetwork parameters before attempting to learn task $t^*$, and $\Delta\theta_{mf}$ is the candidate parameter change computed by the optimizer (Adam). Our base learner is the 6-layer CNN. Unlike von Oswald et al. (2020), we wish to retain the assumption of the rest of the paper where we do not require conditioning inputs; hence we do not use any separate conditioning input (i.e. task embeddings), and use the test-time input as the primary argument, i.e. $\mathcal{I} \equiv \mathcal{X}$. As such, the lack of use of task embeddings is one of the primary changes to our hypernetworks baseline based on von Oswald et al. (2020). In terms of parameter storage, only the weights of the hypernetwork (and not individual weights of points in the CPS) would need to be stored. As we only adapt von Oswald et al. (2020)'s hypernetwork implementation, we use this as our primary baseline, and recommend comparison to other baselines listed in von Oswald et al. (2020).

Starting from a random initialization, we train the hypernetwork first on CIFAR10 (this can be interpreted as re-initializing the network with pre-trained weights, which based on findings in Neyshabur et al. (2020) may indicate that subsequent task parameters may reside in a shared low-loss basin), and store these parameters $\theta_{mf,t}$ in a parameter set $\{\theta_{mf,t}\}^T$. Then we train on a subsequent CIFAR100 task with the Eqt. 1 loss function, where we compute loss w.r.t. the inputs and current timestep's parameters, change in loss w.r.t. a previous task between using the proposed parameters and the parameters last updated at that task (stored in the parameter set; this is enumerated for all past tasks in sequence), and the distance between the current parameters and all prior parameters. Unlike our prior (multi-task) implementation, (i) we minimize the cosine distance between a current task's parameters against prior parameters sequentially, not in parallel (i.e. the subspace end-points are sequentially fixed in the parameter space, we cannot dynamically move the subspace towards a different region), (ii) we do not gain visibility to all task parameters at once, and (iii) we are computing distance w.r.t. multi-task parameters (i.e. each task's parameters is applicable to its own and prior tasks).

**No conditioning inputs** Adaptation without conditioning inputs is a more difficult scenario; though it is used in adapt model parameters when the target distribution changes, it is not commonly used when the source distribution is constant (i.e. not used in adversarial/backdoor defenses). Detecting when shift has occurred is a major implementation hurdle in most work tackling domain shift and task shift. Conditioning inputs are not used w.r.t. a single test-time distribution (adversarial/backdoor attacks), but are usually provided in meta/continual learning, or domain adaptation. Many implementations use support sets in domain adaptation (Ganin et al., 2016b; Hoffman et al., 2017; Peng et al., 2019; Sun et al., 2015) and meta/continual learning (Finn et al., 2017) literature. The implementations assume a task/domain $T_i$ is sampled from a task/domain distribution $p(T)$ associated with a dataset $D_i$, from which we sample a support set $D_i^s = (X_i^s, Y_i^s)$ and a query set $D_i^q = (X_i^q, Y_i^q)$, where a meta learner computes the optimal model parameters $\theta$ with respect to $D_i^s$ in order to predict accurately on $D_i^q$. The support set distribution approximates the query set distribution here. Other than support/query sets, conditioning inputs can manifest as known/seen embeddings: Cheung et al. (2019) learns a set of parameters for each of the $K$ tasks, where these parameters are stored in superposition with each other. The task-specific models are accessed using task-specific "context" information $C_k$ that dynamically "routes" an input towards a specific model retrieved from this superposition. The context vectors $C_k$ are the headers for each task, and they are all orthogonal to each other, hence making no assumption of a metric that can connect or predict these vectors on new or unseen tasks. While there is work that aims to perform unsupervised domain adaptation (Ganin et al., 2016b; Zhou et al., 2021), domain generalization (Arpit et al., 2021; Cha et al., 2021) and unsupervised meta learning (Hsu et al., 2019; Khodadadeh et al., 2019) without explicit conditioning inputs, there is a gap between their performance in an unsupervised setting compared to methods in supervised or settings with conditioning inputs. To exemplify the difficulty of inference without conditioning inputs at test-time, we can refer to the gap between correctly detecting near-OOD (inputs that are near but not i.i.d to the input distribution) and far-OOD (inputs that are far from the input distribution) inputs as being outliers. For far-OOD, an AUROC close to 99% is attainable on CIFAR-100 (in) vs SVHN (out) (Sastry & Oore, 2020). However, the AUROC for near-OOD is around 85% for CIFAR-100 (in) vs CIFAR-10 (out) (Zhang et al., 2020). In other modalities, such as genomics, the AUROC of near-OOD detection can fall to 66% (Ren et al., 2019). Thus, though removing the conditioning input assumption limits the maximum adaptation performance, it is a practical and realistic assumption we retain in our evaluation.

### A.2.2 DISTRIBUTION SHIFTS

**Training regime** Given that a large number of models are trained to evaluate this work (i.e. 3 models are trained in parallel per CPS configuration across all results), except for Figure 2 and Appendix A.3.1 which were trained to the full 5000 epochs, all models trained in this work (including CPS and baselines) are early-stopped when training loss reaches 1.0 (test set accuracy ranging from 55-65%). Though further convergence is possible, we make note of this computational constraint, and recommend evaluating the results in the context of clean test set accuracy presented in tables/figures. All train-time procedures use a seed of 1. All test-time procedures use a seed of 100. The exception to both is any use of Random-BadNet (backdoor perturbations used in CPS, backdoor attack, random permutations and labels), where the seed is the index of perturbation set starting from 1. The defender uses a train-test split of 80-20%. For backdoor attack settings, the attacker uses a traintime-runtime split of 80-20%; this means the attacker contributes 80% of their private dataset (of which $p, \varepsilon = 0.4$ is poisoned) to the defender's dataset, and we evaluate the backdoor attack on the remaining 20% at test-time (of which 1.0 is poisoned). All experiments were performed with respect to the constraints of $1\times$ NVIDIA GeForce RTX 3070.

**Models** In-line with work in loss landscape analysis (Fort et al., 2020), we opt to use the minimum model required to fit/generalize a model to a test set. In-line with the implementation of Snoek et al. (2015) (to ensure the minimum CNN capcity to fit CIFAR100 test sets), we train each CNN with a cross entropy loss function and SGD optimizer for the following depth and width variations: 3-layer [16, 16, 16], 6-layer [16, 16, 16, 32, 32, 32], 9-layer [16, 16, 16, 32, 32, 32, 32, 32, 32], 6-layer-wide [256, 256, 256, 512, 512, 512].

**Adversarial Attack** We use the Projected Gradient Descent (PGD) attack (Madry et al., 2018). With this attack method, adversarial perturbation rate $\varepsilon_a = 0.1$. can sufficiently bring down the attack success rate comparable and similar to that of $\varepsilon_a = 1.0$. Hence, we scale the perturbation against an upper limit 1.0 in our experiments, i.e. $\varepsilon'_a = 0.4 \times \varepsilon_a$. We computes perturbations with respect to the gradients of the defender's model (a more pessimistic, white-box attack).

**Backdoor Attack & Random Permutations** As we would like different trigger perturbations and target poison labels, such as when sampling 100 non-identical random permutations/labels or constructing 3 backdoored sets for CPS training, we implement Random-BadNet (Datta & Shadbolt, 2022), a variant of the baseline dirty-label backdoor attack algorithm, BadNet (Gu et al., 2019). Alike to BadNet and many existing backdoor implementations in literature, Random-BadNet constructs a static trigger. Instead of a single square in the corner as in the case of BadNet, Random-BadNet generates randomized pixels to return unique trigger patterns. Attackers need to specify a set of inputs mapped to target poisoned labels $\{X_i : Y_i^{\text{poison}}\} \in D_i$ to specify the intended label classification, backdoor perturbation rate $\varepsilon_i$ to specify the proportion of an input to be perturbed, and the poison rate $p_i = \frac{|X^{\text{poison}}|}{|X^{\text{clean}}| + |X^{\text{poison}}|}$ to specify the proportion of the private dataset to contain backdoored inputs, to return $X^{\text{poison}} = b_i(X_i, Y_i^{\text{poison}}, \varepsilon_i, p_i)$. In this work, the target poison labels are randomly-sampled, the perturbation rate and poison rate are $p, \varepsilon = 0.4$.

**Stylization** We use the Adaptive Instance Normalization (AdaIN) stylization method (Huang & Belongie, 2017), which is a standard method to stylize datasets such as stylized-ImageNet (Geirhos et al., 2019). Dataset stylization is considered as texture shift or domain shift in different literature. We randomly sample a distinct (non-repeating) style for each attacker. $\alpha$ is the degree of stylization to apply; 1.0 means 100% stylization, 0% means no stylization; we set $\alpha = 0.5$. We follow the implementation in Huang & Belongie (2017) and Geirhos et al. (2019) and stylize CIFAR-10 with the Paintings by Numbers style dataset. We adapt the method for our attack, by first randomly sampling a distinct set of styles for each attacker, and stylizing each attacker's sub-dataset before the insertion of backdoor or adversarial perturbations. This shift also contributes to the realistic scenario that different agents may have shifted datasets given heterogenous sources.

**Rotation** At train-time we rotate the $N$ sets according to their respective index out of $N$: [90, 0, 270, 180, 45, 135, 61, 315, 60, 315, 20, 75]. These rotation degrees were hand-picked and intended to be diverse and varied. With seed 3407, At test-time, we randomly-sampled 100 seeded rotations: [20, 47, 77, 109, 304, 87, 304, 254, 146, 94, 98, 39, 306, 114, 267, 42, 231, 120, 40, 339, 352, 14, 264, 288, 203, 175, 308, 355, 324, 76, 213, 209, 167, 4, 170, 234, 120, 87, 43, 337, 300, 358, 29, 237, 107, 62, 84, 95, 9, 327, 203, 331, 1, 27, 59, 122, 52, 294, 64, 128, 263, 39, 141, 291, 25, 39, 176, 79, 104,

243, 265, 166, 270, 113, 23, 65, 297, 19, 196, 134, 119, 169, 42, 178, 250, 253, 276, 354, 291, 298, 20, 0, 343, 263, 164, 246, 217, 184, 163, 98].

**Task Interpolation** We evaluate interpolated tasks by linearly interpolating a set of tasks: for a set of tasks $\{X\}^N$ where each task $X$ is a set of inputs $x$ (and the number of inputs per task is identical, i.e. $\{\{x\}^M\}^N$), we linearly-interpolate between indexed inputs across a task set to return an interpolated task: $\hat{\mathbf{x}} = \sum_i^N \sum_j^M \alpha_i x_{i,j}$. In this work, we only interpolate between task sets with identical coarse labels (label-shared). We do not interpolate batches, meaning we do not take a subset of one batch in a task and merge it with another subset of another batch in a task (i.e. all inputs are unperturbed, but each interpolated task contains distinct subsets of different task). We do not implement this setting, as we present results on multi-task solutions, which are evaluated on multiple unperturbed tasks with a single parameter point-estimate, thus an interpolated batch setting is already evaluatable.

We linearly-space the range [0,1] into 5 segments (returning $5^3$ different tasks). We ignore the [0,0,0] case, as this results in a black image set, containing no identifiable task-specific features, and thus likely to return random labels - we leave this case out of our evaluation to filter out this noise. For each interpolated image, we clip the colour values per pixel in the range [0, 255]. We keep the [1,1,1] case as, even though it could be all white theoretically, it would only be all white if each and every pixel exceeded 255; in practice (for example, as shown in Appendix A.2.1, pixels may contain colour channel values whose weighted sum per channel may still be less than 255.

We do not interpolate batches, meaning we do not take a subset of one batch in a task and merge it with another subset of another batch in a task (i.e. all inputs are unperturbed, but each interpolated task contains distinct subsets of different task). We do not implement this setting, as we present results on multi-task solutions, which are evaluated on multiple unperturbed tasks with a single parameter point-estimate, thus an interpolated batch setting is already evaluatable.

Appendix A.2.1: Samples of interpolated images. The label of the variations of interpolated image across tasks is 4. The task end-points are: 3 tasks, 3 × same coarse label, train-time task set.

**Single Test-time Distribution Dataset** We use the CIFAR10 dataset, which contains 10 classes, 60,0000 inputs, and 3 colour channels (Krizhevsky, 2009). For test-time perturbations, we evaluate on 100 different examples and tabulate the mean and standard deviation. For train-time perturbations (backdoor), we insert a single backdoor trigger into the defender set before CPS initiated.

**Multiple Test-time Distributions Dataset** We use CIFAR100 dataset, a common baseline for meta/continual learning settings (Krizhevsky, 2009). Similar to CIFAR10 except in that it contains 100 classes containing 600 images each. There are 500 training images and 100 testing images per class. The 100 classes in the CIFAR100 are grouped into 20 superclasses. Each image comes with a "fine" label (the class to which it belongs) and a "coarse" label (the superclass to which it belongs). There are at most 5 tasks of the same coarse label (given there are 5 fine labels per class) per coarse label set. For 5-label-per-set tasks, there are at most 5 different coarse label sets available (for 10-label-per-set tasks, there are at most 2). To evaluate the use of CPS-regularization in hypernetworks for continual learning, the dataset is used in-line with the Split-CIFAR10/100 setting as described in Zenke et al. (2017) and von Oswald et al. (2020). *N-label-set per task* refers to the number of coarse labels contained/mapped in the task; 5(10)-label-set per task means we have 5(10) coarse labels mapped within the task, while 2 × 5(10)-label-set per task means we have 5(10) coarse labels mapped within the task but of 2 different fine label sets. *Distinctively-different coarse labels (per task)* is a distinction from *different coarse labels (of the same set)*, where the former has a unique coarse label set assigned per task (no shared coarse labels across the tasks), while the latter has a

different set of tasks that share coarse labels (but different fine labels). We provide a breakdown of the labels (both coarse and fine) used per task; each task (as a self-contained training set/dataloader) is indicated by the square brackets [].

- (Appendix A.3.8) Seen tasks (3 tasks, 3 × same coarse label, train-time task set): [4 (aquatic mammals → beaver), 1 (fish → aquarium fish), 54 (flowers → orchids), 9 (food → containers), 0 (fruit and vegetables → apples)], [30 (aquatic mammals → dolphin), 32 (fish → flatfish), 62 (flowers → poppies), 10 (food → bottles), 51 (fruit and vegetables → mushrooms)], [55 (aquatic mammals → otter), 67 (fish → ray), 70 (flowers → roses), 16(food → bowls), 53 (fruit and vegetables → oranges)]

- (Appendix A.3.8) Unseen tasks (2 tasks, 2 × same coarse label but unseen fine label): [72 (aquatic mammals → seal), 73 (fish → shark), 82 (flowers → sunflowers), 28 (food → cups), 57 (fruit and vegetables → pears)], [95 (aquatic mammals → whale), 91 (fish → trout), 92 (flowers → tulips), 61 (food → plates), 83 (fruit and vegetables → sweet peppers)]

- (Appendix A.3.8) Unseen tasks (5 tasks, 1 × 5-label-set per task, 3 × same coarse label but unseen fine label, 2 × distinctly different coarse label): [22 (household electrical devices → clock), 5 (household → furniture), 6 (insects → bee), 3 (large carnivores → bear), 12 (large man-made outdoor things → bridge)], [39 (household electrical devices → computer keyboard), 20 (household → bed), 7(insects → beetle), 42 (large carnivores → leopard), 17 (large man-made outdoor things → castle)], [40 (household electrical devices → lamp), 25 (household → chair), 14 (insects → butterfly), 43 (large carnivores → lion), 37 (large man-made outdoor things → house)], [23 (large natural outdoor scenes → cloud), 15 (large omnivores and herbivores → camel), 34 (medium-sized mammals → fox), 26 (non-insect invertebrates → crab), 2 (people → baby)], [27 (reptiles → crocodile), 36 (small mammals → hamster), 47 (trees → maple), 8 (vehicles 1 → bicycle), 41 (vehicles 2 → lawn-mower)]

- (Appendix A.3.6) 2 tasks, 2 × same coarse label: [4 (aquatic mammals → beaver), 1 (fish → aquarium fish), 54 (flowers → orchids), 9 (food → containers), 0 (fruit and vegetables → apples)], [30 (aquatic mammals → dolphin), 32 (fish → flatfish), 62 (flowers → poppies), 10 (food → bottles), 51 (fruit and vegetables → mushrooms)]

- (Appendix A.3.6) 2 tasks, 2 × different coarse label: [4 (aquatic mammals → beaver), 1 (fish → aquarium fish), 54 (flowers → orchids), 9 (food → containers), 0 (fruit and vegetables → apples)], [22 (household electrical devices → clock), 5 (household → furniture), 6 (insects → bee), 3 (large carnivores → bear), 12 (large man-made outdoor things → bridge)]

- (Appendix A.3.5) 2 tasks (2 × 5-label-set per task, 2 × same coarse label per task, 0 × distinctly different coarse label per task): [4 (aquatic mammals → beaver), 30 (aquatic mammals → dolphin), 1 (fish → aquarium fish), 32 (fish → flatfish), 54 (flowers → orchids), 62 (flowers → poppies), 9 (food → containers), 10 (food → bottles), 0 (fruit and vegetables → apples), 51 (fruit and vegetables → mushrooms)], [55 (aquatic mammals → otter), 72 (aquatic mammals → seal), 67 (fish → ray), 73 (fish → shark), 70 (flowers → roses), 82 (flowers → sunflowers), 16 (food → bowls), 28 (food → cups), 53 (fruit and vegetables → oranges), 57 (fruit and vegetables → pears)]

- (Appendix A.3.5) 3 tasks (2 × 5-label-set per task, 2 × same coarse label per task, 1 × distinctly different coarse label per task): [4 (aquatic mammals → beaver), 30 (aquatic mammals → dolphin), 1 (fish → aquarium fish), 32 (fish → flatfish), 54 (flowers → orchids), 62 (flowers → poppies), 9 (food → containers), 10 (food → bottles), 0 (fruit and vegetables → apples), 51 (fruit and vegetables → mushrooms)], [55 (aquatic mammals → otter), 72 (aquatic mammals → seal), 67 (fish → ray), 73 (fish → shark), 70 (flowers → roses), 82 (flowers → sunflowers), 16 (food → bowls), 28 (food → cups), 53 (fruit and vegetables → oranges), 57 (fruit and vegetables → pears)], [22 (household electrical devices → clock), 39 (household electrical devices → clock), 5 (household → furniture), 20 (household → furniture), 6 (insects → bee), 7 (insects → bee), 3 (large carnivores → bear), 42 (large carnivores → bear), 12 (large man-made outdoor things → bridge), 17 (large man-made outdoor things → bridge)]

- (Appendix A.3.5) 4 tasks (2 × 5-label-set per task, 2 × same coarse label per task, 2 × distinctly different coarse label per task): [4 (aquatic mammals → beaver), 30 (aquatic mammals → dolphin), 1 (fish → aquarium fish), 32 (fish → flatfish), 54 (flowers → orchids), 62 (flowers → poppies), 9 (food → containers), 10 (food → bottles), 0 (fruit and vegetables → apples), 51 (fruit and vegetables → mushrooms)], [55 (aquatic mammals → otter), 72 (aquatic mammals → seal), 67 (fish → ray), 73 (fish → shark), 70 (flowers → roses), 82 (flowers → sunflowers), 16 (food → bowls), 28 (food → cups), 53 (fruit and vegetables → oranges), 57 (fruit and vegetables → pears)], [22 (household electrical devices → clock), 39 (household electrical devices → clock), 5 (household → furniture), 20 (household → furniture), 6 (insects → bee), 7 (insects → bee), 3 (large carnivores → bear), 42 (large carnivores → bear), 12 (large man-made outdoor things → bridge), 17 (large man-made outdoor things → bridge)], [23 (large natural outdoor scenes → cloud), 33 (large natural outdoor scenes → forest), 15 (large omnivores and herbivores → camel), 19 (large omnivores and herbivores → cattle), 34 (medium-sized mammals → fox), 63 (medium-sized mammals → porcupine), 26 (non-insect invertebrates → crab), 45 (non-insect invertebrates → lobster), 2 (people → baby), 11 (people → boy)]

- (Appendix A.3.7) 3 tasks (1 × 5-label-set per task, 3 × same coarse label): [4 (aquatic mammals → beaver), 1 (fish → aquarium fish), 54 (flowers → orchids), 9 (food → containers), 0 (fruit and vegetables → apples)], [30 (aquatic mammals → dolphin), 32 (fish → flatfish), 62 (flowers → poppies), 10 (food → bottles), 51 (fruit and vegetables → mushrooms)], [55 (aquatic mammals → otter), 67 (fish → ray), 70 (flowers → roses), 16(food → bowls), 53 (fruit and vegetables → oranges)]

- (Appendix A.3.7) 3 tasks (1 × 5-label-set per task, 2 × same coarse label, 1 × different coarse label): [4 (aquatic mammals → beaver), 1 (fish → aquarium fish), 54 (flowers → orchids), 9 (food → containers), 0 (fruit and vegetables → apples)], [30 (aquatic mammals → dolphin), 32 (fish → flatfish), 62 (flowers → poppies), 10 (food → bottles), 51 (fruit and vegetables → mushrooms)]], [22 (household electrical devices → clock), 5 (household → furniture), 6 (insects → bee), 3 (large carnivores → bear), 12 (large man-made outdoor things → bridge)]

- (Appendix A.3.7) 3 tasks (1 × 5-label-set per task, 3 × distinctly different coarse label): [4 (aquatic mammals → beaver), 1 (fish → aquarium fish), 54 (flowers → orchids), 9 (food → containers), 0 (fruit and vegetables → apples)], [22 (household electrical devices → clock), 5 (household → furniture), 6 (insects → bee), 3 (large carnivores → bear), 12 (large man-made outdoor things → bridge)], [23 (large natural outdoor scenes → cloud), 15 (large omnivores and herbivores → camel), 34 (medium-sized mammals → fox), 26 (non-insect invertebrates → crab), 2 (people → baby)]

- (Appendix A.3.7) 5 tasks (1 × 5-label-set per task, 5 × same coarse label): [4 (aquatic mammals → beaver), 1 (fish → aquarium fish), 54 (flowers → orchids), 9 (food → containers), 0 (fruit and vegetables → apples)], [30 (aquatic mammals → dolphin), 32 (fish → flatfish), 62 (flowers → poppies), 10 (food → bottles), 51 (fruit and vegetables → mushrooms)], [55 (aquatic mammals → otter), 67 (fish → ray), 70 (flowers → roses), 16(food → bowls), 53 (fruit and vegetables → oranges)], [72 (aquatic mammals → seal), 73 (fish → shark), 82 (flowers → sunflowers), 28 (food → cups), 57 (fruit and vegetables → pears)], [95 (aquatic mammals → whale), 91 (fish → trout), 92 (flowers → tulips), 61 (food → plates), 83 (fruit and vegetables → sweet peppers)]

- (Appendix A.3.7) 5 tasks (1 × 5-label-set per task, 2 × same coarse label, 3 × distinctly different coarse label): [4 (aquatic mammals → beaver), 1 (fish → aquarium fish), 54 (flowers → orchids), 9 (food → containers), 0 (fruit and vegetables → apples)], [30 (aquatic mammals → dolphin), 32 (fish → flatfish), 62 (flowers → poppies), 10 (food → bottles), 51 (fruit and vegetables → mushrooms)]], [22 (household electrical devices → clock), 5 (household → furniture), 6 (insects → bee), 3 (large carnivores → bear), 12 (large man-made outdoor things → bridge)], [23 (large natural outdoor scenes → cloud), 15 (large omnivores and herbivores → camel), 34 (medium-sized mammals → fox), 26 (non-insect invertebrates → crab), 2 (people → baby)], [27 (reptiles → crocodile), 36 (small mammals → hamster), 47 (trees → maple), 8 (vehicles 1 → bicycle), 41 (vehicles 2 → lawn-mower)]

- (Appendix A.3.7) 5 tasks (1 × 5-label-set per task, 2 × same coarse label, 3 × different coarse label of same set): [4 (aquatic mammals → beaver), 1 (fish → aquarium fish), 54 (flowers → orchids), 9 (food → containers), 0 (fruit and vegetables → apples)], [30 (aquatic mammals → dolphin), 32 (fish → flatfish), 62 (flowers → poppies), 10 (food → bottles), 51 (fruit and vegetables → mushrooms)], [22 (household electrical devices → clock), 5 (household → furniture), 6 (insects → bee), 3 (large carnivores → bear), 12 (large man-made outdoor things → bridge)], [39 (household electrical devices → clock), 20 (household → furniture), 7 (insects → bee), 42 (large carnivores → bear), 17 (large man-made outdoor things → bridge)], [40 (household electrical devices → lamp), 25 (household → chair), 14 (insects → butterfly), 43 (large carnivores → lion), 37 (large man-made outdoor things → house)]

## A.3   Supporting Results

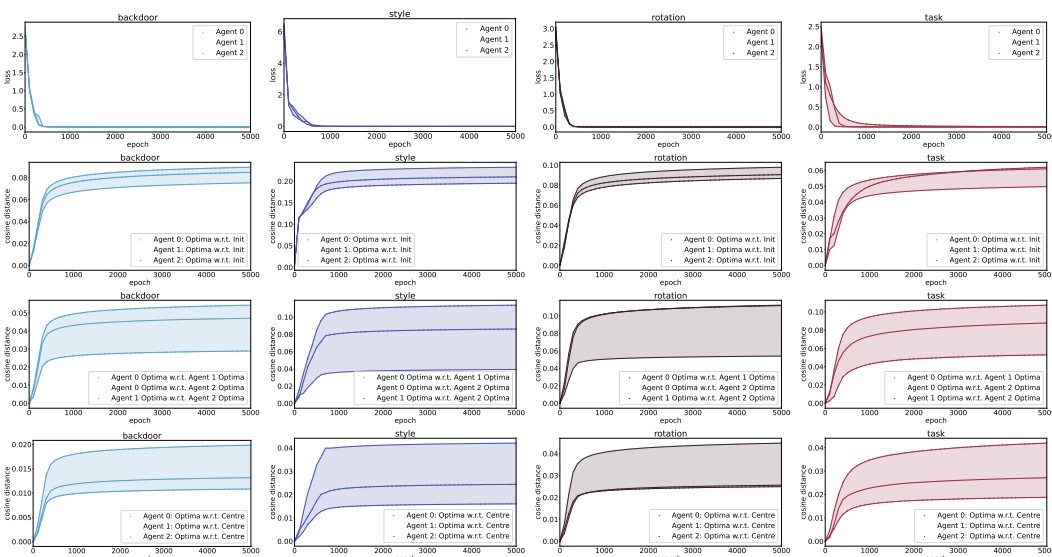

Appendix A.3.1: *Change in parameter subspace dynamics:*  Loss and cosine distance per epoch during training of CPS with 3 train-time distributions ($\beta = 0.0$). Perturbation type of train-time subsets are backdoor (column 1), stylization (column 2), rotation (column 3), and task (column 4).

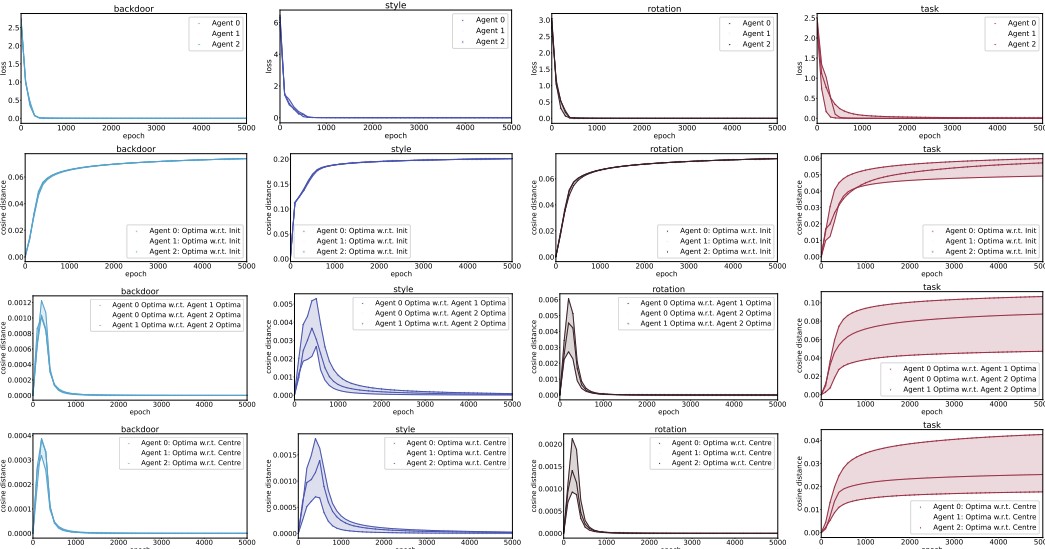

Appendix A.3.2: *Change in parameter subspace dynamics:*  Loss and cosine distance per epoch during training of CPS with 3 train-time distributions ($\beta = 1.0$). Perturbation type of train-time subsets are backdoor (column 1), stylization (column 2), rotation (column 3), and task (column 4).

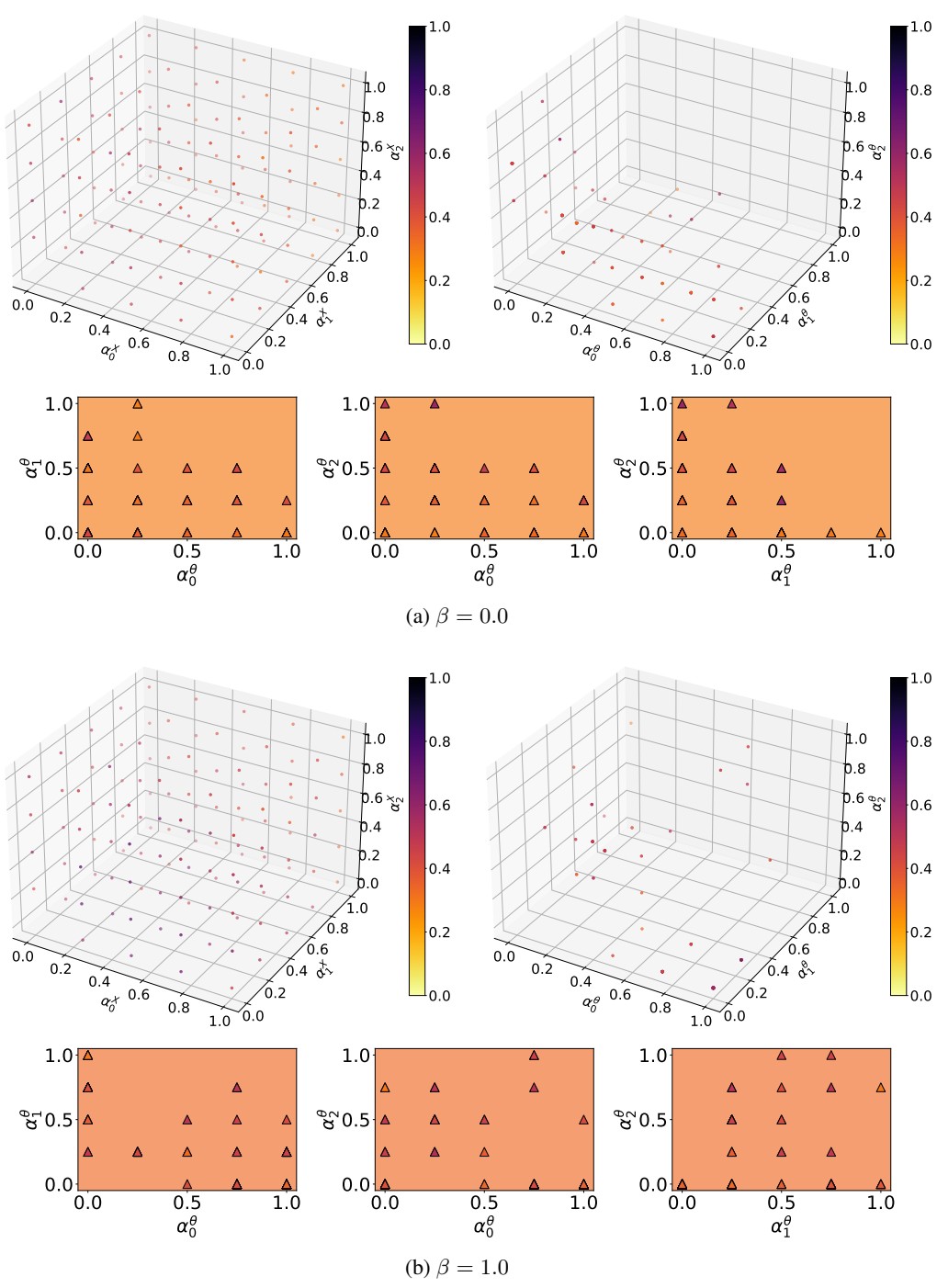

Appendix A.3.3: *Change in parameter subspace dynamics:* 3D plot of interpolated input space (left) and interpolated parameter space (right) for parameter spaces trained with $\beta = 0.0$ (a) and $\beta = 1.0$ (b). The cross-sectional diagrams reflect the corresponding points in 2D. All colours correspond to the color bar in the figure. We only plot the first-instance (i.e. there may be multiple) lowest-loss / maximum-accuracy mappable parameter in the parameter subspace for each interpolated task.

**3-layer CNN**

| Type / # sets | | 2 | 3 | 5 | 10 |
|---|---|---|---|---|---|
| CPS (backdoor) | centre | $29.6 \pm 7.8$ | $55.3 \pm 15.8$ | $49.9 \pm 14.2$ | $49.9 \pm 13.6$ |
| | ens. (mean) | $33.7 \pm 8.7$ | $55.5 \pm 15.7$ | $49.7 \pm 14.4$ | $49.9 \pm 13.7$ |
| | intp. (max) | $55.1 \pm 14.9$ | $57.5 \pm 15.5$ | $52.7 \pm 15.6$ | $55.7 \pm 15.5$ |
| CPS (stylization) | centre | $13.6 \pm 2.0$ | $57.5 \pm 17.8$ | $39.3 \pm 1.8$ | $34.4 \pm 4.9$ |
| | ens. (mean) | $20.9 \pm 3.8$ | $57.3 \pm 18.3$ | $39.8 \pm 2.0$ | $33.8 \pm 4.3$ |
| | intp. (max) | $64.0 \pm 20.2$ | $59.7 \pm 18.2$ | $52.5 \pm 2.8$ | $52.9 \pm 5.0$ |
| CPS (rotation) | centre | $19.4 \pm 0.6$ | $19.0 \pm 1.8$ | $17.2 \pm 2.7$ | $9.9 \pm 3.0$ |
| | ens. (mean) | $19.6 \pm 0.4$ | $19.2 \pm 1.8$ | $17.1 \pm 2.2$ | $15.7 \pm 2.7$ |
| | intp. (max) | $21.7 \pm 0.5$ | $59.6 \pm 17.2$ | $49.7 \pm 1.6$ | $50.5 \pm 5.6$ |

**6-layer CNN**

| Type / # sets | | 2 | 3 | 5 | 10 |
|---|---|---|---|---|---|
| CPS (backdoor) | centre | $60.2 \pm 13.0$ | $48.4 \pm 2.4$ | $50.7 \pm 2.1$ | $52.7 \pm 4.7$ |
| | ens. (mean) | $59.3 \pm 12.8$ | $48.6 \pm 2.9$ | $49.3 \pm 1.7$ | $52.2 \pm 4.7$ |
| | intp. (max) | $61.9 \pm 12.9$ | $51.8 \pm 2.4$ | $52.8 \pm 1.7$ | $56.7 \pm 4.6$ |
| CPS (stylization) | centre | $22.6 \pm 0.8$ | $47.9 \pm 1.1$ | $48.2 \pm 2.2$ | $52.6 \pm 3.9$ |
| | ens. (mean) | $29.6 \pm 1.1$ | $47.4 \pm 1.0$ | $47.7 \pm 2.5$ | $51.8 \pm 3.5$ |
| | intp. (max) | $53.4 \pm 1.4$ | $50.0 \pm 0.8$ | $51.1 \pm 2.9$ | $55.9 \pm 3.5$ |
| CPS (rotation) | centre | $15.9 \pm 0.1$ | $19.4 \pm 1.2$ | $21.2 \pm 3.1$ | $24.8 \pm 3.4$ |
| | ens. (mean) | $17.3 \pm 0.5$ | $19.8 \pm 1.2$ | $20.8 \pm 2.3$ | $24.2 \pm 3.0$ |
| | intp. (max) | $48.1 \pm 2.5$ | $51.4 \pm 1.5$ | $49.5 \pm 4.0$ | $51.3 \pm 3.7$ |

Appendix A.3.4: *CPS vs single test-time distributions:* The relationship between the type of trained perturbations (backdoor / stylization / rotation), number of perturbed sets, and model capacity.

| | | layers $\ell$ | | |
|---|---|---|---|---|
| | | 3 | 6 | 9 |
| 2 tasks (2 × 5-label-set per task, 2 × same coarse label per task, 0 × distinctly different coarse label per task) | centre | $48.0 \pm 12.1$ | $20.0 \pm 1.2$ | $30.1 \pm 3.2$ |
| | ens. (mean) | $68.0 \pm 32.0$ | $24.0 \pm 2.4$ | $35.4 \pm 0.0$ |
| | intp. (max) | $86.6 \pm 9.1$ | $65.7 \pm 34.2$ | $91.0 \pm 9.0$ |
| 3 tasks (2 × 5-label-set per task, 2 × same coarse label per task, 1 × distinctly different coarse label per task) | centre | $29.5 \pm 4.8$ | $20.6 \pm 2.0$ | $20.6 \pm 1.1$ |
| | ens. (mean) | $29.3 \pm 5.3$ | $23.6 \pm 1.7$ | $19.9 \pm 0.8$ |
| | intp. (max) | $83.4 \pm 12.0$ | $74.1 \pm 36.3$ | $53.8 \pm 32.7$ |
| 4 tasks (2 × 5-label-set per task, 2 × same coarse label per task, 2 × distinctly different coarse label per task) | centre | $17.7 \pm 1.9$ | $19.2 \pm 2.1$ | $20.1 \pm 0.6$ |
| | ens. (mean) | $19.6 \pm 0.4$ | $19.4 \pm 0.8$ | $21.8 \pm 1.5$ |
| | intp. (max) | $38.5 \pm 27.5$ | $64.0 \pm 36.2$ | $49.3 \pm 26.7$ |

Appendix A.3.5: *CPS vs multiple test-time distributions:* Model depth against distinct coarse label sets.

**6-layer CNN**

| | | narrow | wide |
|---|---|---|---|
| 2 tasks / same | centre | $17.5 \pm 2.1$ | $37.9 \pm 4.6$ |
| | ens. (mean) | $24.2 \pm 5.6$ | $43.2 \pm 2.2$ |
| | intp. (max) | $62.1 \pm 37.9$ | $82.7 \pm 3.5$ |
| 2 tasks / different | centre | $23.5 \pm 6.9$ | $21.7 \pm 0.8$ |
| | ens. (mean) | $30.9 \pm 10.8$ | $31.0 \pm 11.4$ |
| | intp. (max) | $62.9 \pm 37.1$ | $60.8 \pm 39.2$ |

Appendix A.3.6: *CPS vs multiple test-time distributions:* Tasks (*same/different* coarse labels) against model width.

| | | layers $\ell$ | | |
|---|---|---|---|---|
| | | 3 | 6 | 9 |
| 3 tasks (1 × 5-label-set per task, 3 × same coarse label) | centre | $21.7 \pm 2.2$ | $31.0 \pm 8.4$ | $25.1 \pm 2.2$ |
| | ens. (mean) | $23.3 \pm 4.6$ | $31.0 \pm 4.9$ | $26.8 \pm 1.3$ |
| | intp. (max) | $89.7 \pm 7.0$ | $88.5 \pm 4.5$ | $93.9 \pm 7.8$ |
| 3 tasks (1 × 5-label-set per task, 2 × same coarse label, 1 × different coarse label) | centre | $20.7 \pm 5.2$ | $24.2 \pm 1.4$ | $22.5 \pm 2.7$ |
| | ens. (mean) | $24.3 \pm 9.3$ | $26.9 \pm 3.0$ | $21.2 \pm 1.1$ |
| | intp. (max) | $87.4 \pm 3.1$ | $83.9 \pm 4.5$ | $57.8 \pm 31.4$ |
| 3 tasks (1 × 5-label-set per task, 3 × distinctly different coarse label) | centre | $19.0 \pm 6.5$ | $21.7 \pm 1.2$ | $21.9 \pm 3.8$ |
| | ens. (mean) | $21.2 \pm 6.9$ | $25.0 \pm 3.5$ | $22.4 \pm 1.2$ |
| | intp. (max) | $91.4 \pm 8.3$ | $89.0 \pm 5.6$ | $74.0 \pm 36.7$ |
| 5 tasks (1 × 5-label-set per task, 5 × same coarse label) | centre | $20.2 \pm 1.3$ | $23.2 \pm 6.6$ | $20.2 \pm 1.3$ |
| | ens. (mean) | $20.2 \pm 1.3$ | $26.3 \pm 3.6$ | $21.2 \pm 1.2$ |
| | intp. (max) | $21.8 \pm 0.7$ | $87.7 \pm 24.7$ | $61.1 \pm 32.0$ |
| 5 tasks (1 × 5-label-set per task, 2 × same coarse label, 3 × distinctly different coarse label) | centre | $20.2 \pm 1.3$ | $19.9 \pm 1.5$ | $23.5 \pm 5.4$ |
| | ens. (mean) | $20.2 \pm 1.3$ | $21.4 \pm 1.5$ | $21.6 \pm 2.4$ |
| | intp. (max) | $21.6 \pm 0.5$ | $84.8 \pm 30.3$ | $55.8 \pm 36.3$ |
| 5 tasks (1 × 5-label-set per task, 2 × same coarse label, 3 × different coarse label of same set) | centre | $20.2 \pm 1.3$ | $23.7 \pm 3.5$ | $20.0 \pm 1.9$ |
| | ens. (mean) | $20.2 \pm 1.3$ | $23.2 \pm 1.9$ | $21.0 \pm 2.1$ |
| | intp. (max) | $21.6 \pm 0.5$ | $73.7 \pm 32.3$ | $60.6 \pm 33.0$ |

Appendix A.3.7: *CPS vs multiple test-time distributions:* Varying model depth against varying task label set diversity (different fine labels, different coarse labels).

| | | CPS: 3 tasks, 3 × same coarse label, 6-layer CNN, unique-task solution | CPS: 3 tasks, 3 × same coarse label, 6-layer CNN, multi-task solution | CPS: 2 tasks, 2 × same coarse label, 6-layer-wide CNN, unique-task solution | CPS: 2 tasks, 2 × same coarse label, 6-layer-wide CNN, multi-task solution |
|---|---|---|---|---|---|
| **-** | | | | | |
| Seen tasks (3 tasks, 3 × same coarse label, train-time task set) | centre | $33.9 \pm 5.7$ | $27.2 \pm 0.0$ | $38.9 \pm 4.8$ | $37.2 \pm 0.0$ |
| | ens. (mean) | $31.0 \pm 4.8$ | $28.6 \pm 0.0$ | $36.3 \pm 3.2$ | $35.9 \pm 0.0$ |
| | intp. (max) | $59.4 \pm 9.1$ | $43.3 \pm 0.0$ | $57.2 \pm 8.6$ | $40.6 \pm 0.0$ |
| Unseen tasks (2 tasks, 2 × same coarse label but unseen fine label) | centre | $34.8 \pm 2.7$ | $33.8 \pm 0.0$ | $31.0 \pm 0.2$ | $30.0 \pm 0.0$ |
| | ens. (mean) | $29.4 \pm 0.2$ | $29.3 \pm 0.0$ | $32.2 \pm 3.8$ | $31.3 \pm 0.0$ |
| | intp. (max) | $42.9 \pm 5.8$ | $41.7 \pm 0.0$ | $40.4 \pm 5.4$ | $35.2 \pm 0.0$ |
| Unseen tasks (5 tasks, 1 × 5-label-set per task, 3 × same coarse label but unseen fine label, 2 × distinctly different coarse label) | centre | $18.4 \pm 3.5$ | $18.0 \pm 0.0$ | $23.3 \pm 4.3$ | $23.0 \pm 0.0$ |
| | ens. (mean) | $18.8 \pm 2.4$ | $18.9 \pm 0.0$ | $23.1 \pm 3.1$ | $22.2 \pm 0.0$ |
| | intp. (max) | $30.9 \pm 1.1$ | $28.0 \pm 0.0$ | $30.3 \pm 2.2$ | $32.0 \pm 0.0$ |
| **+ Adversarial Attack** | | | | | |
| Seen tasks (3 tasks, 3 × same coarse label, train-time task set) | centre | $29.4 \pm 6.1$ | $25.0 \pm 0.0$ | $36.7 \pm 3.6$ | $31.7 \pm 0.0$ |
| | ens. (mean) | $30.0 \pm 4.7$ | $27.9 \pm 0.0$ | $36.5 \pm 4.1$ | $32.2 \pm 0.0$ |
| | intp. (max) | $55.6 \pm 5.2$ | $42.8 \pm 0.0$ | $52.8 \pm 5.7$ | $37.8 \pm 0.0$ |
| Unseen tasks (2 tasks, 2 × same coarse label but unseen fine label) | centre | $33.3 \pm 11.7$ | $32.5 \pm 0.0$ | $31.7 \pm 5.0$ | $38.3 \pm 0.0$ |
| | ens. (mean) | $30.9 \pm 6.1$ | $27.1 \pm 0.0$ | $31.8 \pm 0.5$ | $34.4 \pm 0.0$ |
| | intp. (max) | $42.5 \pm 2.5$ | $37.5 \pm 0.0$ | $38.3 \pm 3.3$ | $42.5 \pm 0.0$ |
| Unseen tasks (5 tasks, 1 × 5-label-set per task, 3 × same coarse label but unseen fine label, 2 × distinctly different coarse label) | centre | $15.0 \pm 3.8$ | $19.0 \pm 0.0$ | $21.0 \pm 4.7$ | $20.0 \pm 0.0$ |
| | ens. (mean) | $18.6 \pm 3.1$ | $18.9 \pm 0.0$ | $23.9 \pm 3.1$ | $22.3 \pm 0.0$ |
| | intp. (max) | $25.3 \pm 2.9$ | $22.3 \pm 0.0$ | $33.0 \pm 6.9$ | $29.0 \pm 0.0$ |
| **+ Random Permutations** | | | | | |
| Seen tasks (3 tasks, 3 × same coarse label, train-time task set) | centre | $33.3 \pm 4.9$ | $27.8 \pm 0.0$ | $39.4 \pm 5.5$ | $37.8 \pm 0.0$ |
| | ens. (mean) | $31.6 \pm 4.6$ | $28.7 \pm 0.0$ | $37.2 \pm 3.8$ | $35.6 \pm 0.0$ |
| | intp. (max) | $60.8 \pm 21.7$ | $43.3 \pm 0.0$ | $58.9 \pm 7.5$ | $40.6 \pm 0.0$ |
| Unseen tasks (2 tasks, 2 × same coarse label but unseen fine label) | centre | $34.8 \pm 2.7$ | $33.1 \pm 0.0$ | $31.3 \pm 0.0$ | $29.6 \pm 0.0$ |
| | ens. (mean) | $29.4 \pm 0.0$ | $28.8 \pm 0.0$ | $32.7 \pm 3.1$ | $31.1 \pm 0.0$ |
| | intp. (max) | $42.7 \pm 6.0$ | $41.9 \pm 0.0$ | $41.5 \pm 4.4$ | $35.6 \pm 0.0$ |
| Unseen tasks (5 tasks, 1 × 5-label-set per task, 3 × same coarse label but unseen fine label, 2 × distinctly different coarse label) | centre | $18.5 \pm 3.7$ | $18.3 \pm 0.0$ | $22.9 \pm 4.4$ | $21.7 \pm 0.0$ |
| | ens. (mean) | $18.5 \pm 2.2$ | $19.0 \pm 0.0$ | $23.1 \pm 2.7$ | $22.7 \pm 0.0$ |
| | intp. (max) | $30.8 \pm 1.1$ | $28.0 \pm 0.0$ | $30.3 \pm 2.1$ | $29.3 \pm 0.0$ |
| **+ Stylization** | | | | | |
| Seen tasks (3 tasks, 3 × same coarse label, train-time task set) | centre | $28.3 \pm 11.6$ | $25.0 \pm 0.0$ | $27.2 \pm 0.8$ | $31.7 \pm 0.0$ |
| | ens. (mean) | $27.0 \pm 6.1$ | $25.8 \pm 0.0$ | $27.5 \pm 2.0$ | $30.9 \pm 0.0$ |
| | intp. (max) | $48.3 \pm 5.9$ | $36.7 \pm 0.0$ | $36.7 \pm 6.2$ | $35.6 \pm 0.0$ |
| Unseen tasks (2 tasks, 2 × same coarse label but unseen fine label) | centre | $33.1 \pm 1.5$ | $38.5 \pm 0.0$ | $32.1 \pm 4.2$ | $29.0 \pm 0.0$ |
| | ens. (mean) | $29.5 \pm 0.2$ | $30.1 \pm 0.0$ | $30.3 \pm 2.0$ | $30.3 \pm 0.0$ |
| | intp. (max) | $42.3 \pm 0.6$ | $41.5 \pm 0.0$ | $36.3 \pm 3.3$ | $35.6 \pm 0.0$ |
| Unseen tasks (5 tasks, 1 × 5-label-set per task, 3 × same coarse label but unseen fine label, 2 × distinctly different coarse label) | centre | $19.4 \pm 3.9$ | $16.0 \pm 0.0$ | $23.9 \pm 4.6$ | $19.7 \pm 0.0$ |
| | ens. (mean) | $20.5 \pm 1.5$ | $18.2 \pm 0.0$ | $25.3 \pm 3.2$ | $22.2 \pm 0.0$ |
| | intp. (max) | $32.8 \pm 3.3$ | $31.0 \pm 0.0$ | $32.8 \pm 2.5$ | $33.3 \pm 0.0$ |
| **+ Rotation** | | | | | |
| Seen tasks (3 tasks, 3 × same coarse label, train-time task set) | centre | $33.3 \pm 6.2$ | $26.1 \pm 0.0$ | $32.2 \pm 5.5$ | $32.8 \pm 0.0$ |
| | ens. (mean) | $30.7 \pm 5.1$ | $27.3 \pm 0.0$ | $32.8 \pm 3.9$ | $32.9 \pm 0.0$ |
| | intp. (max) | $54.4 \pm 3.4$ | $37.2 \pm 0.0$ | $52.8 \pm 7.5$ | $35.6 \pm 0.0$ |
| Unseen tasks (2 tasks, 2 × same coarse label but unseen fine label) | centre | $33.5 \pm 4.0$ | $34.2 \pm 0.0$ | $30.4 \pm 0.8$ | $35.0 \pm 0.0$ |
| | ens. (mean) | $28.5 \pm 1.0$ | $29.1 \pm 0.0$ | $31.5 \pm 4.9$ | $34.8 \pm 0.0$ |
| | intp. (max) | $41.3 \pm 7.5$ | $40.8 \pm 0.0$ | $39.6 \pm 6.3$ | $37.9 \pm 0.0$ |
| Unseen tasks (5 tasks, 1 × 5-label-set per task, 3 × same coarse label but unseen fine label, 2 × distinctly different coarse label) | centre | $17.8 \pm 3.2$ | $19.7 \pm 0.0$ | $21.7 \pm 3.1$ | $18.7 \pm 0.0$ |
| | ens. (mean) | $18.4 \pm 2.6$ | $17.9 \pm 0.0$ | $21.5 \pm 1.1$ | $19.6 \pm 0.0$ |
| | intp. (max) | $31.4 \pm 1.5$ | $25.3 \pm 0.0$ | $28.6 \pm 3.3$ | $30.3 \pm 0.0$ |

Appendix A.3.8: *CPS vs multiple test-time distributions:* Evaluating seen/unseen tasks (varying perturbation types) with CPS of varying over-parameterization (number of tasks and model width).

