# OpenReview forum: "Interpolating Compressed Parameter Subspaces"
_ICLR.cc/2023/Conference — Submitted to ICLR 2023_

### Official Review · Reviewer_326H · 2022-10-21

**Confidence:** 4
**Correctness:** 3
**Technical Novelty And Significance:** 2
**Empirical Novelty And Significance:** 2
**Recommendation:** 3

**Clarity, Quality, Novelty And Reproducibility:**

Clarity

This paper is well organized.

Quality

The paper has come technical concerns that need to be addressed.

Novelty

The novelty of this paper is reasonable.

Reproducibility

Good.

**Strength And Weaknesses:**

Strength

+ The study of subspace structure gains more insight to various critical aspects of deep learning.

Weaknesses

- Theorem 2 and its proof in A.1 is confusing. With (9), does it mean ∂L(θi; Xi)/∂θi = ∂L(θn; Xn)/∂θn? θi and θn are trained with different train set from {Di}N, why will SGD update them always in the same direction?
- The main contribution and focus of this paper is "Departing from constructing subspaces w.r.t. a single/unperturbed input distribution, we investigate the construction of subspces w.r.t. multiple perturbed distributions". However, the results in Table 1 show underperforming accuracies comparing with models trained with single input distributions. For example, CPS (backdoor) performs better than Backdoor Adversarial Training, but worse in  Stylization and Rotation. CPS (stylization) and CPS (rotation) perform much worse in Backdoor Attack. The effectiveness of CPS is questionable and inconclusive based on the experiment results.
-"strong capacity for multitask solutions and unseen/distant tasks" is not fully supported by experiment results as Adversarial Attack and Random Permutations get better accuracies, but Stylization and Rotation are worse.
-β is the key parameter to CPS. However, it is not studies sufficiently in the paper. The experiments only include β=0 and β=1.0.
- N + 1 parameters have to be trained, which is computationally expensive even with parallel training.
- Please make the texts in Figure 1 larger.
- Please fix "that can multiple types of distribution shifts concurrently"

**Summary Of The Paper:**

This paper presents a new finding in subspace by constructing a Compressed Parameter Subspaces (CPS). It is found that a
geometric structure representing distance-regularized parameters mapped to a set of train-time distributions can maximize average accuracy over a broad range of distribution shifts.

**Summary Of The Review:**

This paper investigates the interesting filed of subspace and CPS. There are some technical concerns and experiment results are not convincing.

---

### Official Review · Reviewer_jM1a · 2022-10-25

**Confidence:** 4
**Correctness:** 2
**Technical Novelty And Significance:** 3
**Empirical Novelty And Significance:** 3
**Recommendation:** 3

**Clarity, Quality, Novelty And Reproducibility:**

There are many typos and mistakes in the equations. The work is novel in considering the domain shift problem from the point of parameter space.

**Strength And Weaknesses:**

strength：The work propose to tackle multiple types of distribution shift concurrently, and on some cases the performance is better than augmentation, which may declose a new way for enhance the robustness of a model.

Weakness: the method is not real domain generalization because the shift in test data need to be known during training. The method is complicated, because it needs to train a model for every interpolation type with a specific {alpha}. Typos, especially in equations, makes it difficult to follow the main idea. The performance is much lower than augmentation in some cases. The relationship of this work to "parameter subspaces" in the title is not clear.

some concerns are listed,
1.It looks more like a model level information combination method rather than the title claimed “compressed”.
2.Legends in Figure 1 is too small.
3.The definition 1 is confusing, it says that x^ is a interpolation between x0 and x_i^det, but the equation contains only  x_i^det, the definition of i is more confusing.
4.Symbol N has two different meaning is not a wise choice.
5.Bottom of page2, Where does x_i come from?
6.Page 6 on task interpolation, the equation for x^ , is it true that alpha is only related to index i? why?
7.Table1, Why the performance of CPS sometimes are better than augmentation and sometimes much worse than augmentation? Especially for the stylization and rotation cases. For CPS with rotation, the performance on clean test set is destroyed seriously, why?

8.What about CPS when comparing to some domain generalization or domain adaptation methods?

Typos:
1.Page2 “ that can multiple types of”
2.Algorithm 1, there are two D_i s in row10, what’s the difference?
3.Algorithm 1, how to compute the loss “cos()” in row 13?


**Summary Of The Paper:**

This work propose a method to tackle the train-test inconsistent problem, e.g. distribution/domain/task shift exists at test time. The method tries to find a way to linearly combine multi models (obtained from some samples interpolated via train and test data) so that it can adapt to seen tasks as well as unseen interpolated tasks. It’s not a real domain generalization method because the shift in test data need to be known during training.

**Summary Of The Review:**

The work needs to be polished more seriously, a lot of mistakes in equations make it difficult to understand the main idea. I think it’s not ready for publishing.

---

### Official Review · Reviewer_haBW · 2022-10-27

**Confidence:** 3
**Correctness:** 3
**Technical Novelty And Significance:** 3
**Empirical Novelty And Significance:** 3
**Recommendation:** 5

**Clarity, Quality, Novelty And Reproducibility:**

The writing and technical points are clear, but in my opinion, the overall idea is not that novel although this paper delivers several good points. However, this paper has value in constructing a robust compressed parameter subspace. I cannot access their supplementary material so I cannot judge the reproducibility.

**Strength And Weaknesses:**

- S1) A new method robust to distribution shifts
- S2) A theoretical framework to have a robust parameter
- S3) A simple algorithmic design

- W1) During training, N times larger resources are needed.
- W2) The cosine distance regularization may fail to find the compressed parameter subspace.
- W3) It is unclear whether or not more advanced optimizers do not break the theorems.

**Summary Of The Paper:**

This paper proposes to learn multiple parameters (with a cosine distance regularization to emphasize the diversity of the learned parameters). During a testing period, one can use the interpolated parameter to increase model robustness. The authors also prove a couple of important theorems that i) a compressed parameter subspace can be learned and ii) the interpolated parameter is a good selection. An extensive set of experimental results are introduced.

**Summary Of The Review:**

I think this paper has value since it introduces a new approach to having a robust parameter set. But I have several concerns below:

- Q1) During training, N times larger resources are needed.
- Q2) The cosine distance regularization may fail to find the compressed parameter subspace.
- Q3) It is unclear whether or not more advanced optimizers do not break the theorems (due to the momentum of Adam).
- Q4) Why didn't you compare with other ensemble-based baselines? Since your method also internally build an ensemble of models, it is fair to compare with them.
- Q5) CIFAR-10/100 are too restrictive environments. Do you have more results in other datasets/tasks?
- Q6) Why don't we train sequentially to have a better-compressed parameter subspace? Is training in parallel really the best?

---

### Decision · Program_Chairs · 2023-01-20

**Decision:**

Reject

**Justification For Why Not Higher Score:**

Many parts of the paper contains technical errors that make a good understanding of the method and assessment of its soundness hard to perform.

**Justification For Why Not Lower Score:**

N/A

**Metareview: Summary, Strengths And Weaknesses:**

The paper proposes a method for linearly combining multiple models learned from interpolated data so that the combination better adapts to unseen interpolated tasks.
While the idea could be interesting on its own, many concerns were raised by the reviewers about the clarity and quality of the presentation. In particular, there appear to be many mistakes /unclear steps in the equations that make the whole idea/approach difficult to understand as pointed out by Reviewers jM1a and 326H.